# HiMoE-VLA: Hierarchical Mixture-of-Experts for Generalist Vision–Language–Action Policies

## Abstract

The development of foundation models for embodied intelligence critically depends on access to large-scale, high-quality robot demonstration data. Recent approaches have sought to address this challenge by training on large collections of heterogeneous robotic datasets. However, unlike vision or language data, robotic demonstrations exhibit substantial heterogeneity across embodiments and action spaces as well as other prominent variations such as senor configurations and action control frequencies. The lack of explicit designs for handling such heterogeneity causes existing methods to struggle with integrating diverse factors, thereby limiting their generalization and leading to degraded performance when transferred to new settings. In this paper, we present HiMoE-VLA, a novel vision–language–action (VLA) framework tailored to effectively handle diverse robotic data with heterogeneity. Specifically, we introduce a Hierarchical Mixture-of-Experts (HiMoE) architecture for the action module which adaptively handles multiple sources of heterogeneity across layers and gradually abstracts them into shared knowledge representations. Through extensive experimentation with simulation benchmarks and real-world robotic platforms, HiMoE-VLA demonstrates a consistent performance boost over existing VLA baselines, achieving higher accuracy and robust generalization across diverse robots and action spaces.

## 1 Introduction

The success of vision–language models (VLMs) in capturing rich multimodal representations (Beyer et al., 2024; Touvron et al., 2023; Achiam et al., 2023; Jiang et al., 2023) has motivated their extension into robotics, giving rise to vision–language–action (VLA) models that integrate perception, instruction understanding, and control. By leveraging multimodal inputs, VLA models (Brohan et al., 2022; Stone et al., 2023) can map visual observations and language instructions into executable robot actions (Zitkovich et al., 2023; Kim et al., 2024; Team et al., 2024b; Black et al., 2024; Reuss et al., 2025; Cheang et al., 2025). With the increasing availability of large-scale robotic datasets (O'Neill et al., 2024; Khazatsky et al., 2024), these models have recently demonstrated encouraging progress in manipulation, marking an important step toward robotic foundation models.

Compared to the relative uniformity of textual and visual data in VLM, current VLA models face a fundamental challenge: large-scale robotic datasets are inherently heterogeneous from different aspects. Robots differ in embodiment, action space, state representation, and control frequency; observations vary across number of sensors, viewpoints, and environments; and even when identical tasks are collected in the same environment, variations in teleoperation styles, such as operator speed, can introduce additional heterogeneity. This diversity makes knowledge transfer across datasets and embodiments particularly difficult. As a result, a central and pressing question for the field is how to learn a generalizable foundation model for robotics from such highly heterogeneous robotic data.

Recent methods (Li et al., 2024; Qu et al., 2025; Kim et al., 2025; Liu et al., 2024) pre-train on large-scale datasets such as the Open X-Embodiment (OXE) dataset (O'Neill et al., 2024) and subsequently fine-tune on specific target domains in pursuit of robotic foundation models. Although this paradigm has yielded encouraging results, it still lacks principled designs to effectively handle

data heterogeneity and diversity. As a result, they often struggle to integrate diverse data, leading to limited generalization and inefficient knowledge transfer.

In this paper, we introduce HiMoE-VLA, a vision–language–action (VLA) framework grounded in a Hierarchical Mixture-of-Experts (HiMoE) architecture, designed to enable robust knowledge transfer across diverse robotic datasets. The framework integrates two complementary components: a pretrained vision–language model (VLM) that processes visual and text inputs, and a hierarchical MoE module that operates on robot states and noisy action signals. Considering that data from different action spaces are largely non-transferable, directly mixing this source of heterogeneity with other variations often leads to integration difficulties (as empirically demonstrated in Table 6 (b)). To address this challenge, we propose a hierarchical expert structure composed of three complementary components. At the boundary layers, the Action-Space MoE (AS-MoE) specializes in handling discrepancies between action spaces (e.g., joint-angle-space versus end-effector–space control). Adjacent to it, the Heterogeneity-Balancing MoE (HB-MoE) adaptively processes broader sources of variability, such as embodiment-specific kinematics and sensor configurations. At the middle layers, a dense transformer block consolidates these heterogeneous signals into shared representations, thereby enabling effective cross-domain generalization.

To further enhance this hierarchical abstraction process, we introduce two targeted regularizations. Action-Space Regularization (AS-Reg), implemented as a contrastive objective, sharpens expert specialization over different action spaces. Heterogeneity-Balancing Regularization (HB-Reg) guides experts to progressively abstract broader sources of variability into unified knowledge. Additionally, we employ a flow-matching loss to effectively model multimodal action distributions. Together, these objectives constitute the unified training signal of HiMoE-VLA, promoting both robust knowledge transfer and principled expert specialization within the framework.

We pre-train HiMoE-VLA on the OXE (O'Neill et al., 2024) dataset as well as the open-source ALOHA (Fu et al., 2024; Zhao et al., 2023; Liu et al., 2024) dataset, covering diverse embodiments, action spaces, state representations, and tasks. Building on this pre-training, we fine-tune and evaluate HiMoE-VLA across multiple challenging benchmarks, including CALVIN (Mees et al., 2022) and LIBERO (Liu et al., 2023a), as well as on two distinct robot platforms, xArm and ALOHA. Extensive experiments demonstrate that HiMoE-VLA achieves state-of-the-art performance, significantly surpassing existing VLA baselines in both success rates and generalization. Notably, our model exhibits strong generalization to unseen objects and environments, as well as robust adaptation to new robots and tasks, underscoring the effectiveness of our design.

Our contributions are summarized as follows:

- We propose a new Vision–Language–Action framework targeted at handling diverse robotic data with heterogeneity - ranging from action and state spaces to embodiments and sensor configurations - into shared knowledge representations, thus facilitating effective cross-domain transfer.

- We introduce a hierarchical Mixture-of-Expert architecture with an Action-Space MoE (AS-MoE) and a Heterogeneity-Balancing MoE (HB-MoE), supported by targeted regularizations. The AS-MoE addresses discrepancies across action spaces, while the HB-MoE abstracts broader variability into shared knowledge.

- Our model achieves better performance than previous VLA approaches across both simulation benchmarks and real-world single-arm and dual-arm robot platforms, exhibiting quick adaptation to new robots and tasks and effective generalization to unseen objects and environments

## 2 RELATED WORK

**Vision-Language-Action Models.** Rapid progress of large language models (LLMs) (Achiam et al., 2023; Touvron et al., 2023; Team et al., 2024a) and vision-language models (VLMs) (Abdin et al., 2024; Beyer et al., 2024) has spurred the development of vision-language-action (VLA) models that couple pretrained VLMs with robotic action generation. Representative approaches include RT-2 (Zitkovich et al., 2023) and OpenVLA (Kim et al., 2024), which discretize actions into tokens, RoboFlamingo (Li et al., 2023), which predicts continuous actions, and UniVLA (Bu et al., 2025) and Pi0 (Black et al., 2024), which incorporate action-aware objectives and multiview

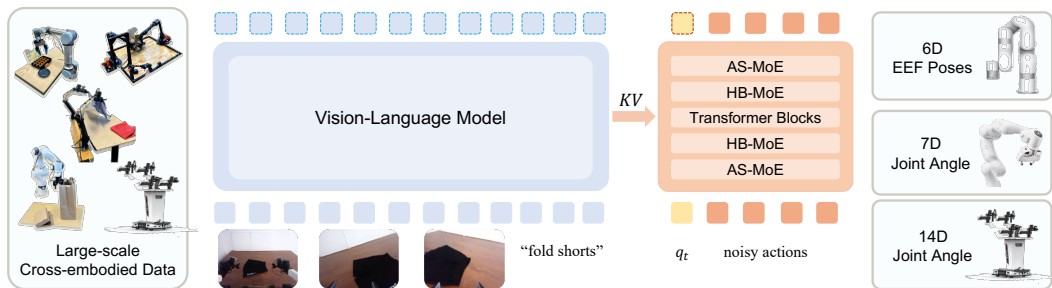

Figure 1: Overview of HiMoE-VLA. The left blue part illustrates the VLM backbone initialized from PaliGemma (Beyer et al., 2024), and the right orange part depicts our proposed action module with a novel Hierarchical Mixture-of-Experts (HiMoE), which is responsible for processing different robot states and noisy actions and generating final action outputs.

inputs. In parallel, video-pretrained policies (Wu et al., 2023; Cheang et al., 2024) exploit Internet-scale videos to learn visuomotor representations without explicit action supervision. Despite these advances, most VLAs overlook the intrinsic heterogeneity of robotic data, including action spaces (e.g., joint-angle vs. end-effector control) and embodiments, which limits their robustness. Recent efforts attempt to address this: RDT-1B (Liu et al., 2024) introduces a unified action space for bimanual manipulation but lacks architectural mechanisms to handle heterogeneity within the same action space, while HPT (Wang et al., 2024a) employs dataset-specific stems and heads to align diverse inputs, at the cost of limiting transfer across datasets. Our work differs by introducing a hierarchical MoE design that explicitly disentangles action-space discrepancies and broader heterogeneity, while consolidating them into shared knowledge representations.

**Mixture of Experts.** Mixture-of-Experts (MoE) architectures were originally proposed to improve scalability by activating only a subset of parameters per input, achieving sparse computation without sacrificing model capacity. This idea has been widely adopted in LLMs (Fedus et al., 2022; Lepikhin et al., 2020), and later extended to vision (Riquelme et al., 2021) and diffusion models (Fei et al., 2024). The most common routing strategy is top-$k$ token routing, where each input token is dynamically assigned to a subset of experts. Various extensions have been proposed to improve routing efficiency and load balancing, such as hashing-based routing (Roller et al., 2021), dynamic expert activation (Guo et al., 2024; Wang et al., 2024b), and regularization-based balancing losses (Dai et al., 2024). Compared with prior MoE designs, our hierarchical organization places action-space experts at shallow layers and heterogeneity-balancing experts at deeper layers, interleaved with Transformer blocks. This enables specialization over fine-grained action variations while progressively consolidating broader sources of heterogeneity into shared knowledge representations.

## 3 METHOD

### 3.1 PROBLEM FORMULATION

Our objective is to develop a generalist vision–language–action (VLA) model that enables robots with different embodiments (e.g., single-arm and dual-arm manipulators) to execute diverse tasks conditioned on multimodal inputs. Specifically, at each time step $t$, the model is given a *language instruction* $l$ and multimodal observations consisting of robot proprioception $q_t$ and RGB images $o_t$, and it outputs a sequence of future actions $A_t = [a_t, a_{t+1}, \ldots, a_{t+H-1}]$ over a prediction horizon $H$. Formally, policy $\pi$ can be expressed as:

$$\pi : (l, q_t, o_t) \mapsto A_t,$$

where $q_t$ denotes the proprioceptive state of the robot (e.g., joint positions or end-effector states), and the language instruction $l$ represents the task description expressed as a sequence of tokens. Visual observation $o_t$ is defined as: $o_t = [I_t^1, \ldots, I_t^n]$, where $I_t^i$ denotes the $i$-th RGB image (normally $i$ ranges from 1 to 3).

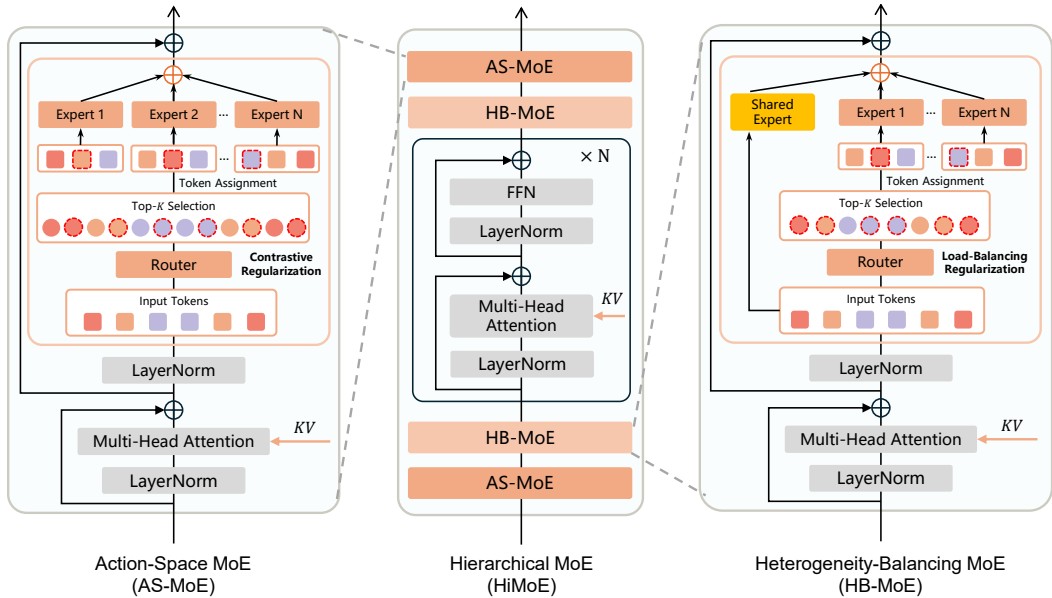

Figure 2: Detailed structure of the Hierarchical Mixture-of-Experts (HiMoE). The architecture follows a layered hierarchy: AS-MoE modules at the boundaries specialize in action-space variations, adjacent HB-MoE modules address broader heterogeneity, and the central Transformer blocks serve as shared layers for cross-domain knowledge integration.

The action sequence $A_t$ is represented as a chunk of low-level robot control signals, where each $a_t$ can correspond to either end-effector deltas or joint angle commands, depending on the embodiment. This chunking formulation allows the model to generate temporally consistent actions that capture fine-grained manipulation dynamics.

## 3.2 NETWORK ARCHITECTURE

In this section, we describe the architecture of the HiMoE-VLA model, as illustrated in Fig. 1, which integrates a pre-trained vision–language backbone (Beyer et al., 2024)) with a dedicated action expert to enable policy learning from multimodal inputs. The model is trained with flow-matching (Lipman et al., 2022) loss for action generation, following recent advances in diffusion-based policy learning. At each time step, the policy takes as input the robot's proprioceptive state, a noised action vector, and cross-attended image–text tokens from the VLM backbone, and produces a denoised sequence of future actions. In the following, we elaborate on each module in detail.

### 3.2.1 VISION-LANGUAGE MODULE

Our VLM adopts the PaliGemma (Beyer et al., 2024) model, identical to that used in $\pi_0$ (Black et al., 2024). PaliGemma combines a SigLIP (Zhai et al., 2023) vision encoder with a Gemma (Team et al., 2024a) language model to produce semantically aligned vision–language representations from input images and language instructions.

We extract intermediate key–value (KV) representations from the language model layers and feed them to the action expert for cross-attention with proprioception and action tokens (see Appendix C for details), which provides stronger conditioning than using only the final layer. At inference time, we employ a KV cache to reuse previously computed representations, substantially accelerating rollout without degrading performance.

### 3.2.2 ACTION MODULE WITH HIERARCHICAL MOE

On the action side, we propose a Hierarchical Mixture-of-Experts (HiMoE) architecture, referred to as the action expert, to process the robot's proprioceptive state together with the noised action

sequences. Both inputs are first projected into a unified vector representation, where different action spaces (e.g., joint-angle-based or end-effector–based control) are consistently assigned to fixed positions within the vector. These unified vectors are normalized to zero mean and unit variance across the dataset, and subsequently transformed by lightweight MLPs before being passed into the HiMoE.

The HiMoE itself is composed of two key expert modules—Action-Space MoE (AS-MoE) and Heterogeneity-Balancing MoE (HB-MoE)—interleaved with standard Transformer blocks (see Fig. 2 for details). The AS-MoE operates at shallow layers to specialize in action-space–specific processing, ensuring that variations such as joint-based versus end-effector–based control are effectively captured. The HB-MoE, in contrast, functions at the adjacent layers to progressively abstract heterogeneous factors and balance representation learning across diverse embodiments, thereby consolidating the information into shared knowledge.

At each layer, the expert outputs are fused with intermediate key–value (KV) representations extracted from the PaLI-Gemma backbone, enabling the model to integrate low-level visual cues with high-level semantic information throughout the hierarchy. This layer-wise fusion provides rich contextual conditioning: shallow layers achieve effective specialization, while deeper layers promote stronger generalization and transfer across tasks and embodiments. Finally, the fused representations are used to generate denoised action chunks under the flow-matching (Lipman et al., 2022) training objective.

## 3.3 TRAINING OBJECTIVE

The training objective of HiMoE-VLA consists of three components: a flow-matching loss for learning action distributions, an *Action-Space Regularization* (AS-Reg) to enhance expert specialization in the AS-MoE, and a *Heterogeneity-Balancing Regularization* (HB-Reg) to encourage balanced abstraction in the HB-MoE. The overall objective is given by:

$$\mathcal{L} = \mathcal{L}_{\text{flow}} + \lambda_{\text{AS}} \, \mathcal{L}_{\text{AS}} + \lambda_{\text{HB}} \, \mathcal{L}_{\text{HB}}, \tag{1}$$

where $\lambda_{\text{AS}}$ and $\lambda_{\text{HB}}$ control the relative contributions of the two regularization terms. Below, we elaborate on each loss in detail.

**Flow-Matching Loss.** We adopt the flow-matching objective (Lipman et al., 2022) to model the conditional distribution of action sequences, as it provides a more stable and efficient alternative to traditional diffusion training. Given an action chunk $A_t = [a_t, a_{t+1}, \ldots, a_{t+H-1}]$, flow matching defines a continuous-time trajectory that transports a noise distribution to the target action distribution. Specifically, we define perturbed actions as:

$$A_t^\tau = \tau A_t + (1 - \tau)\epsilon, \quad \epsilon \sim \mathcal{N}(0, I), \quad \tau \in [0, 1], \tag{2}$$

where $\tau$ is the flow-matching timestep. The model then learns a vector field $v_\theta$ that predicts the denoising direction:

$$\mathcal{L}_{\text{flow}} = \mathbb{E}_{\tau, A_t, \epsilon} \left[ \left\| v_\theta(A_t^\tau, \tau, o_t, l, q_t) - (\epsilon - A_t) \right\|_2^2 \right], \tag{3}$$

where $o_t$ denotes the visual observation, $l$ the language instruction, and $q_t$ the proprioceptive state. During training, $\tau$ is sampled from a Beta distribution, following practices in recent work such as Black et al. (2024), to emphasize noisier steps and thereby improve robustness. At inference time, future actions are generated by integrating the learned vector field from $\tau = 0$ to $\tau = 1$, starting from Gaussian noise.

**Action-Space Regularization (AS-Reg).** The **AS-MoE**, located at shallow layers of the H-MoE, is designed to capture fine-grained variations in action spaces, such as differences between joint-based and end-effector–based control. To reinforce this specialization, we introduce an *Action-Space Regularization* (AS-Reg) based on a contrastive objective. Let $u \in \{1, \ldots, U\}$ index tokens in the input sequence. For each token $u$, we treat pairs of experts $(i, j)$ assigned to the same action-space token as positive pairs, while pairs $(i, k)$ with $k \neq j$ are considered negatives. Denote by $h_{i,u}$ the

Table 1: CALVIN task performance under $D \to D$. Numbers are the average count of consecutively completed tasks for sequence lengths 1–5 (higher is better).

| Method | 1 | 2 | 3 | 4 | 5 | Sum. |
|---|---|---|---|---|---|---|
| Octo | 0.771 | 0.535 | 0.318 | 0.206 | 0.136 | 1.968 |
| OpenVLA | 0.716 | 0.385 | 0.180 | 0.088 | 0.042 | 1.411 |
| RDT-1B | 0.757 | 0.495 | 0.359 | 0.243 | 0.184 | 2.038 |
| DeeR | 0.853 | 0.696 | 0.549 | 0.420 | 0.312 | 2.830 |
| MDT | **0.937** | 0.845 | 0.741 | 0.644 | 0.556 | 3.723 |
| $\pi_0$ | 0.914 | 0.830 | 0.739 | 0.676 | 0.599 | 3.758 |
| HiMoE-VLA | 0.932 | **0.855** | **0.789** | **0.731** | **0.660** | **3.967** |

Table 2: LIBERO task performance across four suites. Numbers denote average success rates (%) across 50 demonstrations per task.

| Method | Spatial | Object | Goal | Long | Sum. |
|---|---|---|---|---|---|
| Diffusion Policy | 78.3 | 92.5 | 68.3 | 50.5 | 72.4 |
| Octo | 78.9 | 85.7 | 84.6 | 51.1 | 75.1 |
| OpenVLA | 84.7 | 88.4 | 79.2 | 53.7 | 76.5 |
| SpatialVLA | 88.2 | 89.9 | 78.6 | 55.5 | 78.1 |
| OpenVLA-OFT | 97.6 | 98.4 | 97.9 | 94.5 | 97.1 |
| UniVLA | 96.5 | 96.8 | 95.6 | 92.0 | 95.2 |
| $\pi_0$ | 96.8 | 98.8 | 95.8 | 85.2 | 94.2 |
| HiMoE-VLA | **98.2** | **99.4** | **98.6** | **94.8** | **97.8** |

score produced by expert $i$ for token $u$. The loss is defined as

$$\mathcal{L}_{\text{AS}} = -\frac{1}{U} \sum_{u=1}^{U} \log \frac{\exp(\text{sim}(h_{i,u}, h_{j,u})/\tau)}{\sum_{k=1}^{N} \exp(\text{sim}(h_{i,u}, h_{k,u})/\tau)}, \tag{4}$$

$$\text{sim}(h_{i,u}, h_{j,u}) = \frac{h_{i,u} \cdot h_{j,u}}{\|h_{i,u}\| \, \|h_{j,u}\|}, \tag{5}$$

where $\tau$ is a temperature parameter, $N$ is the number of experts, and $\text{sim}(\cdot, \cdot)$ denotes cosine similarity. By encouraging agreement among experts routed to the same action-space tokens while reducing similarity to others, this objective guides AS-MoE experts toward targeted specialization, ensuring that action-space heterogeneity is effectively captured at shallow layers.

**Heterogeneity-Balancing Regularization (HB-Reg).** The **HB-MoE**, in contrast, functions in deeper layers to progressively abstract broader sources of heterogeneity—spanning robot embodiments, sensor configurations, and scene variations—and to consolidate them into shared knowledge. To support this role, we introduce *Heterogeneity-Balancing Regularization* (HB-Reg).

Let $N$ denote the number of experts, $K$ the number of routed experts per token (top-$K$ gating), $U$ the number of tokens in the sequence, and $s_{i,u} \in [0,1]$ the gating score assigned to expert $i$ for the $u$-th token. After top-$K$ selection, we define a binary routing indicator

$$r_{i,u} = \mathbb{1}\{\text{token } u \text{ is routed to expert } i\}.$$

The (empirical) routing frequency and the expected routing probability for expert $i$ are defined as

$$f_i = \frac{1}{KU} \sum_{u=1}^{U} r_{i,u}, \quad P_i = \frac{1}{U} \sum_{u=1}^{U} s_{i,u}. \tag{6}$$

The heterogeneity-balancing loss is then defined as

$$\mathcal{L}_{\text{HB}} = \sum_{i=1}^{N} f_i \, P_i. \tag{7}$$

This objective ensures that the expected routing probability ($P_i$) and the realized routing frequency ($f_i$) are aligned, thus distributing heterogeneous inputs more evenly across experts. In doing so, HB-Reg prevents expert underutilization and promotes balanced abstraction at deeper layers, enabling the HB-MoE to consolidate diverse information into generalizable shared representations.

In summary, AS-Reg drives **specialization** in the AS-MoE for capturing action-space differences at shallow layers, while HB-Reg enforces **balancing** in the HB-MoE for integrating heterogeneous factors at deeper layers. Together with the flow-matching loss, these objectives enable HiMoE-VLA to learn expressive and transferable policies from highly diverse robotic data.

## 4 EXPERIMENTS

**Pre-training Dataset.** We pre-train HiMoE-VLA on a large-scale mixture of the Open X-Embodiment (OXE) subset (O'Neill et al., 2024) (22.5M frames) and publicly available Aloha

Table 3: Real-world evaluation on the XArm7 robot across three single-arm manipulation tasks: "Fruit-to-Plate", "Cup-in-Cup" and "Block-on-Block". Each task is decomposed into sub-stages (Pick/Place, Pick/Insert, Pick/Stack), and success rates are reported for each stage with the overall average across all tasks.

| Method | Fruit-to-Plate | | Cup-in-Cup | | Block-on-Block | | Task (All) |
|---|---|---|---|---|---|---|---|
| | Pick | Place | Pick | Insert | Pick | Stack | Sum. |
| Octo-Base | 31.3 | 18.8 | 33.3 | 16.7 | 16.7 | 0.0 | 19.3 |
| OpenVLA | 37.5 | 25.0 | 27.8 | 16.7 | 22.2 | 0.0 | 21.2 |
| CogACT | 65.6 | 59.4 | 77.8 | 63.9 | 69.4 | 33.3 | 61.5 |
| $\pi_0$ | 68.8 | 62.5 | 77.8 | 61.1 | 72.2 | 33.3 | 62.5 |
| HiMoE-VLA | **81.3** | **75.0** | **88.9** | **72.2** | **83.3** | **50.0** | **75.0** |

Table 4: Real-world evaluation on Aloha dual-arm robot across three manipulation tasks: "Fold-Shorts", "Handover" and "Scoop". Each task is decomposed into fine-grained sub-stages (e.g., Grasp, Transfer, Place, Pour), and success rates are reported for each stage with the overall average.

| Method | Cup-Handover | | Scoop | | | Fold-Shorts | | Task (All) |
|---|---|---|---|---|---|---|---|---|
| | Grasp | Transfer | Place | Scoop | Pour | Once | Twice | Sum. |
| ACT | 40.0 | 0.0 | 73.3 | 6.6 | 0.0 | 20.0 | 6.6 | 20.9 |
| RDT-1B | 66.6 | 13.3 | 93.3 | 40.0 | 20.0 | 53.3 | 46.6 | 47.5 |
| $\pi_0$ | **80.0** | 13.3 | 93.3 | 46.6 | 26.6 | 66.6 | 53.3 | 54.2 |
| HiMoE-VLA | **80.0** | **26.6** | **100.0** | **53.3** | **40.0** | **80.0** | **66.6** | **63.7** |

datasets (Liu et al., 2024; Zhao et al., 2023; Fu et al., 2024) (1.6M frames), totaling 24.1M frames. This combination provides diverse embodiments, action spaces, and tasks, enabling effective cross-domain learning. More details are provided in the Appendix B.1

**Implementation Details.** HiMoE-VLA (4B parameters) is trained end-to-end on 16 A100 GPUs with DeepSpeed optimization. The model consumes third-person and wrist-mounted camera views, along with unified state–action vectors for both single- and dual-arm settings. All heterogeneous actions and states are mapped into a fixed 24-dimensional vector, consisting of 8-dimensional end-effector actions and 16-dimensional joint angles. For states, a validity mask is concatenated to indicate which segments are active; actions are not masked but zero-padded if a particular type is absent. The MoE design uses $N = 32$ experts with top-$k = 4$, and the auxiliary regularization coefficients are set following best practices. For the MoE gating mechanism, each token's hidden state is fed into a linear projection to compute the expert logits. These logits are normalized via a standard softmax without temperature scaling, producing a distribution over experts. The gate selects the top-$k$ experts based on these scores, and the selected probabilities are renormalized so that their weights sum to one. This design ensures stable routing and well-scaled mixture coefficients throughout training. More details are provided in Appendix C.

## 4.1 SIMULATION EXPERIMENTS

**Experiment setup.** We evaluate HiMoE-VLA on two widely used simulation benchmarks: CALVIN (Mees et al., 2022) and LIBERO (Liu et al., 2023a). CALVIN benchmarks instruction-conditioned, long-horizon tabletop manipulation with a Franka Panda arm. We adopt the challenging D→D setting, training on a limited subset of demonstrations and evaluating on held-out instructions, with comparisons against strong baselines including Octo, OpenVLA, RDT-1B, DeeR, MDT, and $\pi_0$.

LIBERO is a simulation suite for lifelong learning and generalization, spanning four complementary task suites—Spatial, Object, Goal, and Long—each with 10 tasks. We follow standard preprocessing and evaluation protocols, comparing against baselines such as Diffusion Policy, Octo, OpenVLA, SpatialVLA, OpenVLA-OFT, UniVLA, and $\pi_0$.

Further dataset statistics, preprocessing details, and fine-tuning protocols are provided in the Appendix B.2

Table 5: Real-world generalization evaluation on single-arm (XArm7) and dual-arm (Aloha) tasks under two scenarios: *Distractor Objects* (unseen distractors) and *Novel Objects* (previously unseen items). Results highlight each method's generalization ability beyond the training distribution.

| Method | Single-Arm | | | Dual-Arm | | |
|---|---|---|---|---|---|---|
| | Distractor | Novel Obj. | Sum. | Distractor | Novel Obj. | Sum. |
| OpenVLA | 19.4 | 15.6 | 17.6 | - | - | - |
| CogACT | 52.8 | 50.0 | 51.5 | - | - | - |
| RDT-1B | - | - | - | 28.9 | 26.7 | 27.8 |
| $\pi_0$ | 58.3 | 53.1 | 55.9 | 40.0 | 26.7 | 33.4 |
| HiMoE-VLA | **69.4** | **65.6** | **67.6** | **53.3** | **46.7** | **50.0** |

**Results.** Table 1 reports CALVIN performance under the D→D setting. HiMoE-VLA achieves the best overall performance, completing an average of 3.94 tasks consecutively, surpassing all baselines. While MDT slightly outperforms HiMoE-VLA on the first task (0.937 vs. 0.932), HiMoE-VLA consistently yields higher success rates from the second task onward, showing stronger robustness in long-horizon execution. Compared to $\pi_0$ (3.76), which is among the strongest baselines, HiMoE-VLA improves by +0.18, and compared to MDT (3.72), by + 0.21. The gap is even larger against earlier methods such as DeeR (2.83), RDT-1B (2.04), Octo (1.97), and OpenVLA (1.41). These results highlight HiMoE-VLA's ability to maintain reliable performance as task sequences grow longer, validating its effectiveness for instruction-conditioned manipulation under limited data.

Table 2 presents results on the four LIBERO task suites. HiMoE-VLA achieves the highest overall average score of 97.8%, outperforming strong generalist baselines such as UniVLA (95.2%) and $\pi_0$ (94.2%). Compared to OpenVLA-OFT, the previous state-of-the-art (97.1%), HiMoE-VLA delivers consistent gains across all four suites: +0.6% on Spatial, +1.0% on Object, +0.7% on Goal, and +0.3% on Long. These results establish HiMoE-VLA as the new SOTA on LIBERO, demonstrating robust generalization across diverse manipulation tasks, including long-horizon planning.

## 4.2 REAL-WORLD EXPERIMENTS

We evaluate our model on two real-world robots: xArm7 single-arm and Aloha dual-arm robots.

**Experiment setup.** For the xArm7 (single-arm, 7-DoF with a 1-DoF gripper), we evaluate three manipulation tasks: (1) *Fruit-to-Plate* — placing fruits (apple, orange) onto colored plates (blue, pink); (2) *Cup-in-Cup* — inserting one colored cup (red, yellow, blue) into another; and (3) *Block-on-Block* — stacking one colored block onto another of a different color. Each task is further decomposed into sub-stages (e.g., Pick/Place, Pick/Insert, Pick/Stack). We collect a total of 320 teleoperated demonstrations: 80 for *Fruit-to-Plate*, 120 for *Cup-in-Cup*, and 120 for *Block-on-Block*, with 20 demonstrations per configuration. For evaluation, *Fruit-to-Plate* uses 4 settings with 4 trials each, while *Cup-in-Cup* and *Block-on-Block* each use 6 settings with 3 trials each. We additionally assess generalization with two tests: (1) introducing distractor objects such as an unseen pomegranate or green cup, and (2) novel-object tasks such as placing fruit on a purple plate not seen during training.

For the Aloha (dual-arm, 14-DoF), we evaluate three tasks: (1) *Fold-Shorts* — folding a pair of shorts with 50 teleoperated demonstrations; (2) *Cup-Handover* — the right arm grasps a colored cup (red, yellow, blue) and hands it to the left arm, which places it on a plate (180 demonstrations total); and (3) *Scoop* — the left arm positions a bowl and the right arm scoops materials (mung beans, black rice, sticky rice) into it (120 demonstrations total). Altogether, 350 demonstrations are collected. Evaluation includes 15 trials for *Fold-Shorts*, 5 trials per color for *Cup-Handover*, and 5 trials per material type for *Scoop*. For generalization, we test (1) distractor objects such as bananas or green apples in *Scoop*, and (2) novel shorts in *Fold-Shorts*.

**Results.** On the xArm7, HiMoE-VLA achieves the best overall average success rate of 75.0%, outperforming strong baselines such as $\pi_0$ (62.5%) and CogACT (61.5%). Gains are consistent across sub-stages, with particularly notable improvements on the challenging *Block-on-Block* task, where HiMoE-VLA reaches 50.0% success in the stacking stage compared to 33.3% for $\pi_0$ and CogACT. In the generalization tests (Table 5), HiMoE-VLA achieves 67.6% average success, again

Figure 3: Qualitative examples of real-world executions on (left) the single-arm xArm7 and (right) the dual-arm Aloha robot. The snapshots cover representative stages across tasks such as *Fruit-to-Plate*, *Block-on-Block*, *Cup-Handover*, and *Scoop*.

Table 6: Ablation studies on HiMoE-VLA. (a) Effect of initialization and pretraining. (b) Comparison with alternative methods for handling heterogeneous action spaces.

| (a) Init. & Pretrain. | | | | | | |
|---|---|---|---|---|---|---|
| **Setting** | **1** | **2** | **3** | **4** | **5** | **Sum.** |
| w/o init | 0.917 | 0.832 | 0.753 | 0.698 | 0.627 | 3.827 |
| w/o pretrain | 0.928 | 0.845 | 0.752 | 0.686 | 0.615 | 3.826 |
| Full | **0.932** | **0.855** | **0.789** | **0.731** | **0.660** | **3.967** |

| (b) The results for various methods of handling heterogeneity. | | | | | | |
|---|---|---|---|---|---|---|
| **Setting** | **1** | **2** | **3** | **4** | **5** | **Sum.** |
| Separate Heads | 0.914 | 0.833 | 0.753 | 0.696 | 0.631 | 3.827 |
| GR00T-Like | 0.913 | 0.835 | 0.764 | 0.702 | 0.642 | 3.856 |
| HiMoE | **0.943** | **0.864** | **0.797** | **0.734** | **0.674** | **4.012** |

outperforming $\pi_0$ (55.9%) and CogACT (51.5%), demonstrating robustness to unseen distractors (e.g., pomegranate, green cup) and novel objects (e.g., purple plate).

On the Aloha, HiMoE-VLA consistently surpasses $\pi_0$ and RDT-1B across all three tasks (Table 4), with particularly large improvements on *Fold-Shorts* and *Scoop*, highlighting the strength of hierarchical experts in coordinated bimanual manipulation. In generalization evaluations (Table 5), HiMoE-VLA achieves the best overall performance, demonstrating resilience to unseen distractors (e.g., banana, green apple) in *Scoop* and novel shorts in *Fold-Shorts*.

## 4.3 MODEL ANALYSIS AND ABLATIONS

We perform a set of experiments and ablations on the CALVIN benchmark to analyze the contributions of different components in HiMoE-VLA and assess its ability to handle heterogeneous data.

**Effect of Initialization and Pretraining.** Table 6 (a) compares models fine-tuned on CALVIN-D with different initialization strategies. Removing MoE re-initialization during fine-tuning (*w/o init*) slightly degrades performance compared to the full model, while training from scratch without pretrained weights (*w/o pretrain*) leads to a more notable drop. These results highlight the importance of leveraging pretrained representations and carefully initialized experts for effective adaptation in data-scarce regimes.

**Comparison with Other Methods on Handling Heterogeneous Action Spaces.** Table 6 (b) compares HiMoE with two alternatives: (1) separate heads for each action representation and (2) a GR00T-style embodiment indicator. Both baselines require manually defining the number of embodiments. In contrast, HiMoE leverages adaptive expert selection to balance specialization and sharing without architectural changes. To further validate the efficiency of HiMoE, we co-train models from scratch on CALVIN-ABC-EEF and CALVIN-D-Joint. As shown in the table, both variants underperform our method, demonstrating the effectiveness of HiMoE in modeling heterogeneous action data. Such entangled heterogeneity makes it difficult for the gating network to learn consistent routing patterns and negatively affects the convergence of the MoE.

**Evaluation on Ability of HiMoE to Handle Heterogeneous Data.** To directly evaluate how well the model handles heterogeneous action spaces, we compare two settings trained entirely from scratch: (1) training only on CALVIN-D (joint-angle actions), and (2) co-training jointly on CALVIN-ABC (EEF actions) and CALVIN-D (joint-angle actions).

As shown in the table 7, co-training with heterogeneous EEF and joint-angle data severely degrades the performance of $\pi_0$ and our model without MoE, indicating that these methods struggle with

Table 7: Evaluation of heterogeneous action co-training from scratch on CALVIN.

| Method | D (Joint) | ABC (EEF) + D (Joint) |
|---|---|---|
| $\pi_0$ | 3.806 | 3.547 (**-0.259**) |
| Ours w/o MoE | 3.819 | 3.777 (**-0.042**) |
| Full (HiMoE) | 3.826 | 4.012 (**+0.186**) |

Table 9: Ablation on experts $N$ and top-$K$ routing.

| $K$ | $N$ | 1 | 2 | 3 | 4 | 5 | Sum. |
|---|---|---|---|---|---|---|---|
| 2 | 2 | 0.895 | 0.811 | 0.757 | 0.712 | 0.648 | 3.823 |
| | 4 | 0.901 | 0.814 | 0.761 | 0.715 | 0.653 | 3.844 |
| | 8 | 0.910 | 0.827 | 0.768 | 0.722 | 0.669 | 3.896 |
| | 16 | 0.920 | 0.846 | 0.781 | 0.733 | 0.671 | 3.951 |
| 4 | 8 | 0.921 | 0.847 | 0.776 | 0.715 | 0.657 | 3.916 |
| | 16 | 0.923 | 0.846 | 0.774 | 0.729 | **0.682** | 3.954 |
| | 32 | **0.943** | **0.864** | **0.797** | 0.734 | 0.674 | **4.012** |
| | 64 | 0.919 | 0.854 | 0.785 | **0.738** | 0.672 | 3.968 |
| 8 | 16 | 0.911 | 0.773 | 0.637 | 0.546 | 0.458 | 3.325 |
| | 32 | 0.897 | 0.794 | 0.719 | 0.673 | 0.612 | 3.695 |

cross-space interference. In contrast, the full HiMoE model not only avoids negative transfer but also improves when trained with heterogeneous data.

**Role of Hierarchical MoE Components.** Table 8 studies the effect of different MoE configurations when trained jointly on CALVIN-ABC (EEF actions) and CALVIN-D (joint actions). Removing all MoE layers (*w/o MoE*) substantially reduces performance, confirming their central role in

Table 8: Evaluation of Hierarchical MoE components ablations.

| Setting | 1 | 2 | 3 | 4 | 5 | Sum. |
|---|---|---|---|---|---|---|
| w/o MoE | 0.918 | 0.837 | 0.744 | 0.681 | 0.597 | 3.777 |
| Full-HB-MoE | 0.917 | 0.847 | 0.774 | 0.713 | 0.650 | 3.901 |
| w/o AS-MoE | 0.909 | 0.831 | 0.769 | 0.718 | 0.646 | 3.873 |
| w/o HB-MoE | 0.904 | 0.826 | 0.749 | 0.708 | 0.649 | 3.836 |
| w/o Reg | 0.904 | 0.822 | 0.753 | 0.702 | 0.654 | 3.835 |
| Single-MoE+Reg | 0.914 | 0.839 | 0.757 | 0.688 | 0.615 | 3.813 |
| Full | **0.943** | **0.864** | **0.797** | **0.734** | **0.674** | **4.012** |

handling heterogeneous action spaces. Using only the HB-MoE module (*Full-HB-MoE*) or combining HB-MoE with Transformer blocks (*HB-MoE+TB*, equivalent to *w/o AS-MoE*) yields partial improvements but still lags behind the full model. Removing the Heterogeneity-Balancing MoE itself (*w/o HB-MoE*) further degrades performance, showing that this component is essential for capturing the broader variability arising from different action types, embodiments, and sensing configurations. Moreover, eliminating the auxiliary regularizations (*w/o Reg*) also decreases performance, validating the necessity of both Action-Space Regularization and Heterogeneity-Balancing Regularization in guiding expert specialization and abstraction. We additionally evaluate a variant that applies both regularization losses to a single MoE layer (*Single-MoE+Reg*). Interestingly, this configuration performs even worse than the *w/o Reg* baseline, suggesting that forcing a single MoE module to learn all heterogeneous factors simultaneously is highly challenging, even with both regularization losses applied.

**Scaling the Number of Experts.** Table 9 studies the effect of the number of experts ($N$) and the top-$k$ routing strategy ($K$). Increasing $N$ generally improves performance, with the best results at $N = 32$, $K = 4$. Further scaling ($N = 64$) provides diminishing returns, while very high routing widths ($K = 8$) cause instability. These findings indicate that moderate $N$ and sparse routing $K$ yield the best trade-off between specialization and stability.

## 5 CONCLUSION

In this work, we introduced HiMoE-VLA, a vision–language–action framework based on a Hierarchical Mixture-of-Experts architecture. By placing Action-Space MoE at shallow layers and Heterogeneity-Balancing MoE at adjacent layers, interleaved with Transformer blocks, our model captures fine-grained action variations while consolidating heterogeneous factors into shared representations. Combined with flow-matching training and auxiliary regularizations, this design enables effective transfer across diverse embodiments, action and state representations, and tasks. Extensive simulation and real-world experiments demonstrate superior performance and robust generalization of HiMoE-VLA. Looking ahead, we envision extending hierarchical expert architectures to broader embodied intelligence scenarios, like mobile manipulation, multi-robot collaboration, and lifelong adaptation.

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

## A  THE USE OF LARGE LANGUAGE MODELS (LLMS)

We used large language models (LLMs) solely as auxiliary tools for language polishing during the paper writing process. LLMs were not involved in research ideation, dataset construction, method design, experiments, or analysis. All scientific contributions, technical content, and claims in this paper are the responsibility of the authors.

## B  DATASET AND EVALUATION

### B.1  PRETRAINING DATASET

Our pre-training dataset is constructed by combining subsets of Open X-Embodiment (OXE) and publicly available Aloha datasets, yielding a total of 24.1M frames. The detailed data mixture is listed in Table 10.

**OXE dataset.** OXE (O'Neill et al., 2024) aggregates over 1 million real-world trajectories collected from 60 datasets across 22 distinct robot embodiments. Following prior works such as Octo (Team et al., 2024b) and OpenVLA (Kim et al., 2024), we adopt a subset containing 22.5M frames, chosen to balance scale and diversity while ensuring compatibility with our training pipeline. This subset spans a wide range of single-arm robots and tasks, providing strong coverage of heterogeneous embodiments and action spaces.

**Aloha datasets.** To complement OXE, we incorporate demonstrations from three high-quality, publicly available Aloha datasets (Liu et al., 2024; Zhao et al., 2023; Fu et al., 2024), contributing 36.28M frames in total. Compared to OXE, they emphasize coordinated bimanual actions and higher-fidelity manipulation skills, substantially enriching the diversity of our training corpus.

### B.2  EVALUATION BENCHMARKS

**CALVIN benchmark.** CALVIN (Mees et al., 2022) is a benchmark for evaluating instruction-conditioned policies in long-horizon tabletop manipulation tasks using a Franka Panda arm. It comprises 34 tasks spanning from simple pick-and-place to articulated object manipulation. In our experiments, we adopt the challenging $D \rightarrow D$ setting, where policies are trained on a limited subset

Table 10: Our training data mixture using datasets from the Open X-Embodiment dataset (O'Neill et al., 2023) and Aloha dataset (Liu et al., 2024; Zhao et al., 2023; Fu et al., 2024).

| Dataset | Ratio |
|---|---|
| Fractal (Brohan et al., 2022) | 23.9% |
| Kuka (Kalashnikov et al., 2018) | 12.9% |
| Bridge (Ebert et al., 2021; Walke et al., 2023) | 11.9% |
| Taco Play (Rosete-Beas et al., 2022; Mees et al., 2023) | 2.7% |
| Jaco Play (Dass et al., 2023) | 0.4% |
| Berkeley Cable Routing (Luo et al., 2023) | 0.2% |
| Roboturk (Mandlekar et al., 2019) | 2.1% |
| Viola (Zhu et al., 2022a) | 0.9% |
| Berkeley Autolab UR5 (Chen et al.) | 1.1% |
| Toto (Zhou et al., 2023) | 1.9% |
| Stanford Hydra Dataset (Belkhale et al., 2023) | 4.5% |
| Austin Buds Dataset (Zhu et al., 2022b) | 0.2% |
| NYU Franka Play Dataset (Cui et al., 2022) | 0.7% |
| Furniture Bench Dataset (Heo et al., 2023) | 2.5% |
| UCSD Kitchen Dataset (Yan et al., 2023) | <0.1% |
| Austin Sailor Dataset (Nasiriany et al., 2022) | 2.2% |
| Austin Sirius Dataset (Liu et al., 2023b) | 1.8% |
| DLR EDAN Shared Control (Quere et al., 2020) | <0.1% |
| IAMLab CMU Pickup Insert (Saxena et al., 2023) | 0.9% |
| UTAustin Mutex (Shah et al., 2023) | 2.3% |
| Berkeley Fanuc Manipulation (Zhu et al., 2023) | 0.8% |
| CMU Stretch (Mendonca et al., 2023) | 0.2% |
| BC-Z (Jang et al., 2022) | 6.9% |
| FMB Dataset (Luo et al., 2024) | 7.2% |
| DobbE (Shafiullah et al., 2023) | 1.4% |
| Aloha Dataset (Liu et al., 2024; Zhao et al., 2023; Fu et al., 2024) | 10.4% |

of demonstrations from environment D and evaluated on held-out instruction sequences in the same environment. This setup evaluates the model's ability to generalize to novel instruction compositions under restricted data conditions. For fair comparison, we include Octo (Team et al., 2024b), OpenVLA (Kim et al., 2024), RDT-1B (Liu et al., 2024), DeeR (Yue et al., 2024), MDT (Reuss et al., 2024), and $\pi_0$ (Black et al., 2024) as baselines. For DeeR and MDT, we directly report the results from their original papers. For Octo, OpenVLA, RDT-1B, and $\pi_0$, we adopt their released pre-trained weights and perform fine-tuning on CALVIN-D following their official training procedures to ensure fairness and reproducibility.

**LIBERO benchmark.** LIBERO (Liu et al., 2023a) is a simulation suite designed to evaluate life-long learning and generalization in robotic manipulation. It contains four task suites—LIBERO-Spatial, LIBERO-Object, LIBERO-Goal, and LIBERO-Long—each comprising 10 tasks with 50 human-teleoperated demonstrations per task. These suites test complementary aspects of generalization: spatial reasoning (Spatial), object-level transfer (Object), goal-directed adaptability (Goal), and long-horizon planning (Long). Following prior works such as OpenVLA (Kim et al., 2024), we preprocess demonstrations by removing failure cases, standardizing image inputs, and ensuring consistent trajectory formatting. In our experiments, we perform supervised fine-tuning within each task suite using the successful demonstrations and evaluate policies on held-out task episodes. We compare against strong baselines, including Diffusion Policy (Chi et al., 2023), Octo (Team et al., 2024b), OpenVLA (Kim et al., 2024), SpatialVLA (Qu et al., 2025), OpenVLA-OFT (Kim et al., 2025), UniVLA (Bu et al., 2025), and $\pi_0$ (Black et al., 2024), where reported results are either taken directly from their papers or reproduced under their released implementations.

**Real-world xArm7 benchmark.** We conduct real-world evaluations on an xArm7 robot (7-DoF manipulator with a 1-DoF gripper) across three tasks: (1) *Fruit-to-Plate* — placing fruits (apple, orange) onto colored plates (blue, pink), e.g., "Pick up the apple and place it onto the blue plate"; (2) *Cup-in-Cup* — inserting one colored cup (red, yellow, blue) into another, e.g., "Put the yellow

cup into the red cup"; (3) *Block-on-Block* — stacking one colored block onto a differently colored block, e.g., "Place the yellow block on top of the red block." Each task is decomposed into sub-stages (e.g., Pick/Place, Pick/Insert, Pick/Stack) for fine-grained evaluation. We collect 320 teleoperated demonstrations in total: 80 (*Fruit-to-Plate*), 120 (*Cup-in-Cup*), and 120 (*Block-on-Block*), with 20 demonstrations per configuration.

*In-distribution evaluation.* **Fruit-to-Plate**: 4 settings × 4 trials/setting = 16 trials in total, where each "setting" is a fruit–plate pairing from {apple, orange} × {blue, pink}. **Cup-in-Cup**: 6 settings × 3 trials/setting = 18 trials, where each "setting" is an *ordered* inner→outer color pair from {red, yellow, blue} with distinct colors (i.e., $3 \times 2 = 6$ ordered pairs). **Block-on-Block**: 6 settings × 3 trials/setting = 18 trials, where each "setting" is an *ordered* top→bottom color pair (distinct) from {red, yellow, blue}.

*Generalization tests.* (1) **Distractors in *Cup-in-Cup***: 6 settings (the same 6 inner→outer color pairs as above) × 3 trials/setting = 18 trials, with an unseen distractor (e.g., a pomegranate or a green cup) placed in the scene. (2) **Novel objects in *Fruit-to-Plate***: 4 settings × 4 trials/setting = 16 trials. Here, "4×4" means that we test four novel configurations — placing a pomegranate onto a blue plate, a pomegranate onto a pink plate, an apple onto a purple plate, and an orange onto a purple plate — with each configuration repeated for 4 trials.

**Real-world Aloha benchmark.** We further evaluate on the Aloha robot (dual-arm, 14-DoF) with three tasks: (1) *Fold-Shorts* — folding a pair of shorts (50 teleoperated demonstrations), e.g., "Fold black shorts through multiple bimanual folds"; (2) *Cup-Handover* — the right arm grasps a colored cup (red, yellow, blue) and hands it to the left arm to place on a plate (60 demos per color; 180 total), e.g., "Pick up the blue cup, switch hands, and place it on the plate"; (3) *Scoop* — the left arm places a bowl centrally, then the right arm uses a spoon to scoop materials (mung beans, black rice, sticky rice) into the bowl (40 demos per material; 120 total), e.g., "Place the bowl in the middle of the table, then scoop the glutinous rice with a spoon." Altogether, 350 demonstrations are collected.

*In-distribution evaluation.* **Fold-Shorts**: 1 setting × 15 trials = 15 trials. **Cup-Handover**: 3 settings (one per cup color) × 5 trials/setting = 15 trials. **Scoop**: 3 settings (one per material type) × 5 trials/setting = 15 trials.

*Generalization tests.* (1) **Distractors in *Scoop***: 3 settings (the same three material types) × 3 trials/setting = 9 trials, with unseen distractors (e.g., banana or green apple) added to the scene. (2) **Novel garment in *Fold-Shorts***: 1 setting (previously unseen shorts) × 15 trials = 15 trials.

## C   IMPLEMENTATION DETAILS

**Model scale and training setup.** Our proposed HiMoE-VLA model contains approximately 4B parameters and is trained end-to-end on 16 NVIDIA A100 GPUs (40GB each) for 100k steps with a global batch size of 256. Training takes around 4 days with DeepSpeed optimization, and we adopt the LeRobot data-loading framework to ensure efficient and scalable handling of large heterogeneous datasets.

**Input modalities.** The visual encoder consumes one third-person camera view together with two wrist-mounted views. When a view is unavailable in a dataset, the corresponding channel is zero-padded and masked using attention masks, ensuring a consistent input format. For state and action inputs, we construct a unified vector representation that jointly accommodates both joint-angle and end-effector signals. In single-arm demonstrations, the available arm is mapped to the right-arm channel, while the left-arm channel is zero-padded with masks to preserve compatibility with dual-arm settings.

**Mixture-of-Experts design.** We set the number of experts to $N = 32$ with a top-$k$ routing of 8. As shown in Table 6, this configuration consistently outperforms alternative settings in terms of both average performance and stability, striking a favorable balance between model capacity and computational efficiency. To encourage effective expert utilization and hierarchical abstraction, we introduce two auxiliary regularizations: an Action-Space regularization term with coefficient $\lambda_{\text{AS}} = 0.002$, and a Heterogeneity-Balancing regularization term with coefficient $\lambda_{\text{HB}} = 0.001$. These choices follow best practices for balancing specialization and generalization in MoE architectures.

**Optimization and fine-tuning.** We adopt the AdamW optimizer with an initial learning rate of $2.5 \times 10^{-5}$, weight decay of $1 \times 10^{-4}$, and a cosine decay schedule. The learning rate is linearly warmed up for the first 1k steps, followed by exponential decay until 30k steps with a final floor of $2.5 \times 10^{-6}$. For fine-tuning, we adapt the batch size and number of steps to each benchmark. On the CALVIN benchmark, we use a global batch size of 32 for 40k steps. On LIBERO, we fine-tune separately for each of the four task suites: For LIBERO-10, GOAL and OBJECT, we use a batch size of 64 for 40k, 45k, 45k steps respectively; For SPATIAL, we use a batch size of 32 and train for 35k steps. For real-world experiments, we fine-tune on both xARM and ALOHA robots with a batch size of 64 for 50k steps.

**Cross-layer KV integration.** At each transformer layer $l$, HiMoE receives the key–value pairs $\{K_l^{\mathrm{V}}, V_l^{\mathrm{V}}\}$ from the corresponding VLM layer, which are concatenated with the locally computed $\{K_l^{\mathrm{H}}, V_l^{\mathrm{H}}\}$ of the action expert:

$$\tilde{K}_l = \left[K_l^{\mathrm{H}}; K_l^{\mathrm{V}}\right], \qquad \tilde{V}_l = \left[V_l^{\mathrm{H}}; V_l^{\mathrm{V}}\right].$$

The query $Q_l^{\mathrm{H}}$ then attends to the fused representation:

$$\mathrm{Attn}_l(Q_l^{\mathrm{H}}, \tilde{K}_l, \tilde{V}_l) = \mathrm{softmax}\left(\frac{Q_l^{\mathrm{H}} \tilde{K}_l^{\top}}{\sqrt{d_k}}\right) \tilde{V}_l.$$

This design enables each HiMoE layer to directly condition on semantically aligned signals from its VLM counterpart, instead of relying solely on the final-layer representation. During inference, we further employ a KV cache to reuse the VLM's intermediate keys and values, substantially accelerating policy rollout without degrading performance.

## D  ANALYSIS OF EXPERT ROUTING

We evaluated the expert routing behavior on CALVIN Joint, CALVIN EEF, and LIBERO EEF datasets. The expert activation heatmaps for both AS-MoE and HB-MoE are shown in the figures 4, 5.

From Figure 4, we observe that CALVIN EEF and LIBERO EEF exhibit similar expert activation patterns, while CALVIN Joint shows a clearly different distribution, reflecting the differences in action space. From Figure 5, the expert activation patterns for CALVIN Joint and CALVIN EEF are similar, as these datasets share the same environment and observation settings except for the action space, whereas LIBERO EEF has a distinct activation pattern.

These visualizations demonstrate that HiMoE effectively adapts its experts to handle heterogeneity across both action spaces and observations, confirming that the hierarchical MoE design enables specialized routing for different data types.

## E  LIMITATIONS AND FUTURE WORK

Our current implementation feeds all VLM layers' features into HiMoE and performs cross-attention using the full set of hidden-state key–value pairs. This design introduces two practical limitations. First, not all VLM layers necessarily contribute equally to downstream task execution, and treating all layers as equally important may introduce redundancy. Second, the hidden states include tokens from multiple viewpoints and text descriptions, yet the model currently injects them into HiMoE without distinguishing their relative importance. Adaptively filtering or weighing VLM features could further improve the system, and exploring such mechanisms is an important direction for future work.

Another limitation concerns data and model scale. Although our model performs well on the evaluated tasks, its overall scale is still modest relative to large vision–language foundation models. This limitation arises primarily from the limited availability of high-quality, diverse robotics datasets. The robotics datasets we use are much smaller and less diverse compared to the large-scale corpora used to train general VLMs. Increasing the model capacity, incorporating stronger pretrained VLMs, and training on more extensive and varied robotics datasets could further enhance generalization and

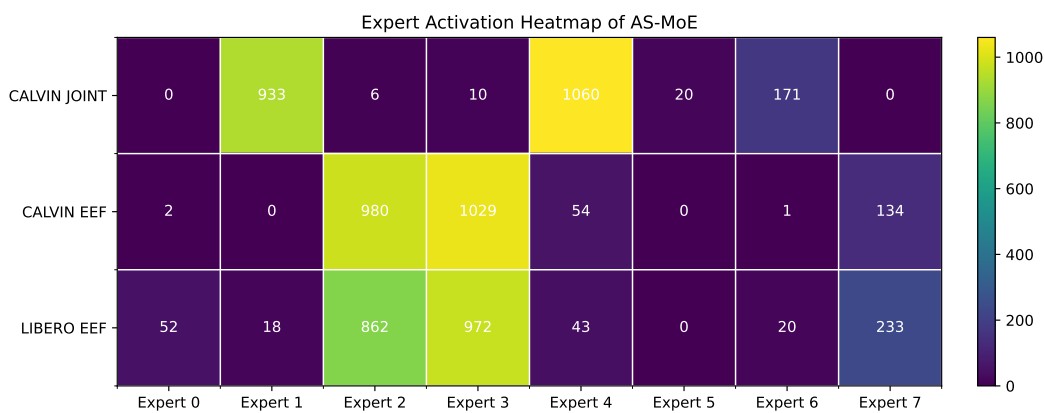

Figure 4: Expert Activation Heatmap of AS-MoE.

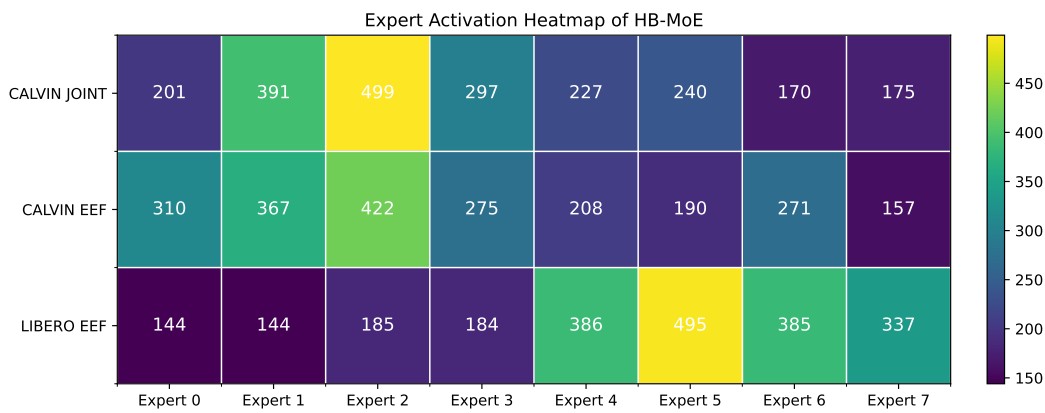

Figure 5: Expert Activation Heatmap of HB-MoE.

robustness. Investigating how model scale interacts with embodiment-specific variability in learning will be an important topic for future research.

## F  MORE VISUALIZATIONS

The virtualization of all tasks are shown in Fig. 6 and Fig. 7.

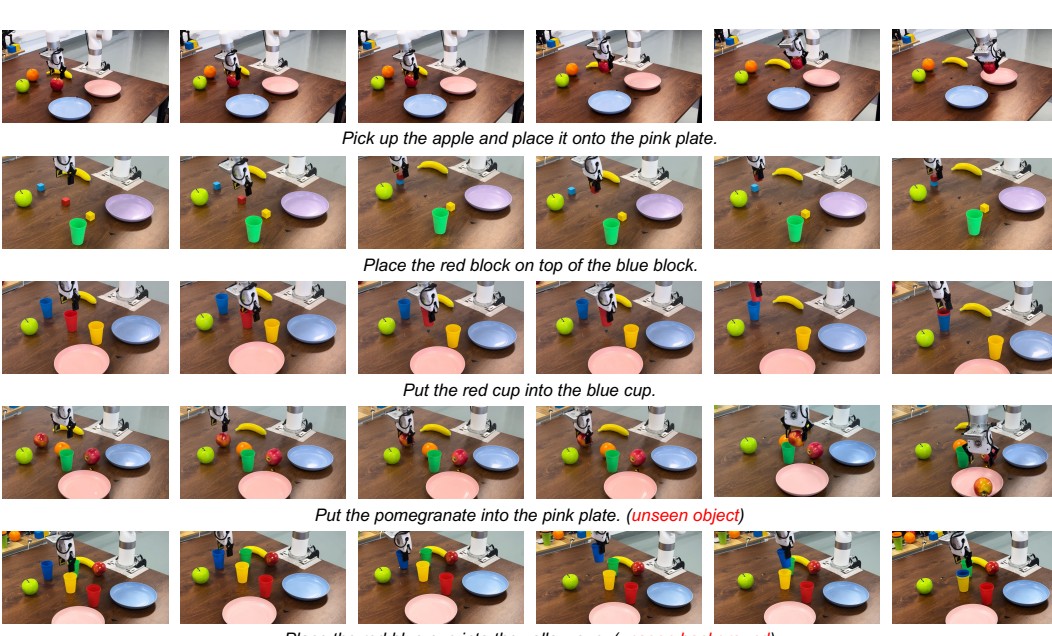

Figure 6: Overview of tasks on singe-arm robot Xarm, including seen tasks and unseen tasks (unseen objects and unseen backgrounds).

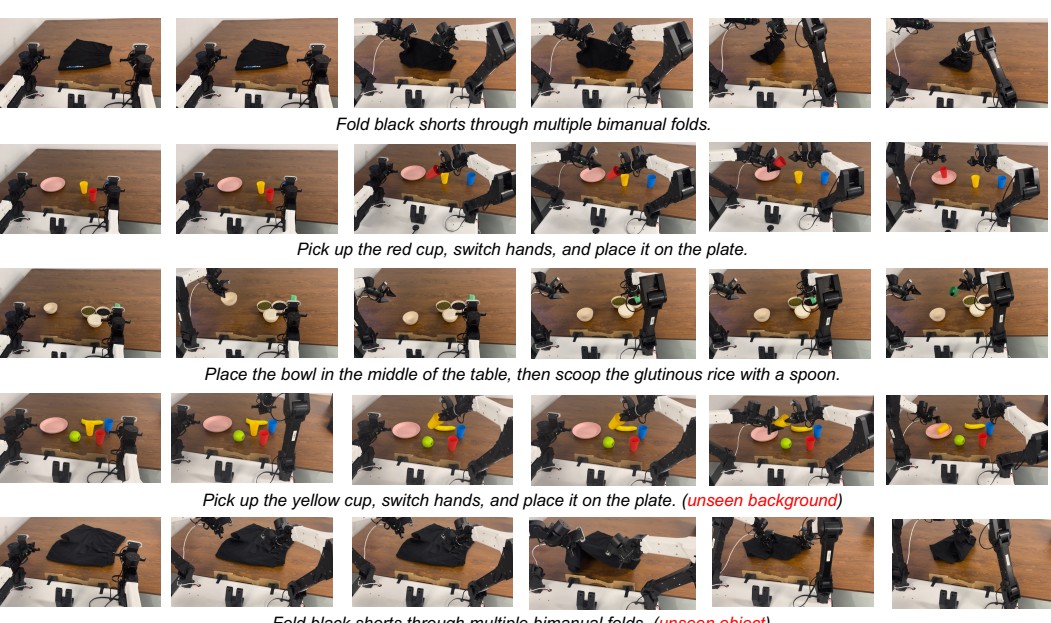

Figure 7: Overview of tasks on dual-arm robot ALOHA, including seen tasks and unseen tasks (unseen objects and unseen backgrounds).

