# OpenReview forum: "HiMoE-VLA: Hierarchical Mixture-of-Experts for Generalist Vision–Language–Action Policies"
_ICLR.cc/2026/Conference — Submitted to ICLR 2026_

### Official Review · Reviewer_gPfg · 2025-10-25

**Soundness:** 2
**Presentation:** 2
**Contribution:** 2
**Rating:** 4
**Confidence:** 3

**Summary:**

To tackle heterogenity in robot learning, this paper presents HiMoE-VLA. Which shares the pipeline with $\pi_0$ but changes the action expert to be a novel architecture named Hierarchical MoE. It contains Action-Space Regularization (AS-Reg) and Heterogeneity-Balancing Regularization (HB-Reg), where experts are dealing with action space heterogeneity and sharing common knowledge separately. Experimental results across 2 simulators and 2 real world robots (one is single arm and the other is dual arm) are sufficient.

**Strengths:**

- The author claims heterogeneity is crucial in robot learning and introduces the Hierarchical MoE method to tackle this.
- Experimental results across 2 simulators and 2 real world robots (one is single arm and the other is dual arm) are sufficient.
- All figures (except for figure 1) are clear and ablation studies are adequate.

**Weaknesses:**

- The performance is poor on CALVIN benchmark, the current SOTA method FLOWER (CoRL 2025) can reach 4,35 in D->D setting, while HiMoE-VLA is only 3.967.
- Figure 1 shows an overview of HiMoE-VLA. However, I cannot see any contributions in this figure, the overall pipeline is the same as $\pi_0$. And the proposed Hierarchical MoE is summarized with only a whole black block. This picture does not bring any useful information about Hierarchical MoE itself.



references:
- [1] FLOWER: Democratizing Generalist Robot Policies with Efficient Vision-Language-Action Flow Policies

**Questions:**

- The test setting is not straightforward. Since the author claims that HiMoE-VLA can tackle with different action space and robotic heterogeneity, then why not train a unified model with OXE, ALOHA and the testing benchmark CALVIN and LIBERO? In appendix C, the author says to fine-tune HiMoE-VLA "separately for each of the four task suite".  More direct experiments are needed to prove that HiMoE-VLA is good at handling heterogeneity.
- What about applying the proposed Hierarchical MoE to other domains? like images or languages. The proposed Hierarchical MoE looks general and can still make sense when action modality is not mentioned here.

---

> ### Author Response · Authors · 2025-11-23
> **Response to Reviewer gPfg (1/2)**
>
> > **W1: The performance is poor on CALVIN benchmark, the current SOTA method FLOWER (CoRL 2025) can reach 4,35 in D->D setting, while HiMoE-VLA is only 3.967.**
>
> We thank the reviewer for the comment. While FLOWER (CoRL 2025) indeed reports remarkable performance on the CALVIN benchmark, we first would like to emphasize that **HiMoE-VLA is already highly competitive** among existing VLA methods. As shown in Table 1 and 2 of our paper, HiMoE-VLA **consistently outperforms** several strong and widely adopted baselines—including **Pi0**, **UniVLA**, **OpenVLA-OFT**, and **Spatial-VLA**. These results, together with the strong evidence in our ablation study, demonstrate that our proposed hierarchical mixture-of-experts design is both **effective and competitive**.
>
> We also note that FLOWER was released in September 2025, essentially *at the same time our work was submitted*. Given the timing, FLOWER and our approach were developed independently.
>
> Last, and most importantly, our method contributes a **fundamentally different and orthogonal technique** compared to FLOWER, i.e., *resolving heterogeneous data through a principled hierarchical MoE framework*. In contrast, *FLOWER focuses on improving VLA efficiency via intermediate-modality fusion and action-specific Global-AdaLN conditioning.* These two lines of work are complementary rather than conflicting.
>
> Our work offers a principled and extensible framework that can be combined with many existing VLA designs—including FLOWER—potentially further improving their performance. Such integration is an exciting direction for future research and does not diminish the strong empirical results demonstrated in our current submission.
>
> [1] π0: A Vision-Language-Action Flow Model for General Robot Control
>
> [2] Unified Vision-Language-Action Model
>
> [3] Fine-Tuning Vision-Language-Action Models: Optimizing Speed and Success
>
> [4] SpatialVLA: Exploring Spatial Representations for Visual-Language-Action Model
>
> [5] FLOWER: Democratizing Generalist Robot Policies with Efficient Vision-Language-Action Flow Policies
>
> > **W2: Figure 1 shows an overview of HiMoE-VLA. However, I cannot see any contributions in this figure, the overall pipeline is the same as . And the proposed Hierarchical MoE is summarized with only a whole black block. This picture does not bring any useful information about Hierarchical MoE itself.**
>
> Thank you for raising this point. First, we would like to clarify the purpose of Figure 1 and its relationship to the technical contributions of our work. Figure 1 is intended as a high-level overview to help readers understand the overall structure and data flow of HiMoE-VLA, *not* as the place where our major methodological contribution is presented. Its role is similar to overview figure in prior VLA works such as $pi_0$ or GR3—providing pipeline and data flow rather than introducing specific contributions. **Accordingly, the method itself should not be identified from the visual layout of Figure 1. Instead, the core technical contribution—our Hierarchical MoE architecture—is fully detailed in Figure 2 and the main text, where the architectural design and innovations are presented.**
>
> Nevertheless, we acknowledge that the original Figure 1 did not sufficiently highlight our contributions, and we wish to thank the reviewer for the insightful feedback. Accordingly, we have revised the figure so that the Hierarchical MoE is no longer represented as a single opaque block. Instead, the updated overview explicitly illustrates its core components in sequence—**AS-MoE → HB-MoE → Transformer blocks → HB-MoE → AS-MoE**—clearly revealing the hierarchical flow and expert interactions. This revision highlights the structure of our design while keeping the figure readable, and it ensures that the core ideas behind HiMoE are now visually communicated even in the overview.
>
> [1] π0: A Vision-Language-Action Flow Model for General Robot Control
>
> [2] GR-3 Technical Report

---

> > ### Author Response · Authors · 2025-11-23
> > **Response to Reviewer gPfg (2/2)**
> >
> > > **Q1: The test setting is not straightforward. Since the author claims that HiMoE-VLA can tackle with different action space and robotic heterogeneity, then why not train a unified model with OXE, ALOHA and the testing benchmark CALVIN and LIBERO? In appendix C, the author says to fine-tune HiMoE-VLA "separately for each of the four task suite". More direct experiments are needed to prove that HiMoE-VLA is good at handling heterogeneity.**
> >
> > The datasets used in testing benchmarks such as CALVIN and LIBERO are relatively small. As shown in prior works [1–3], existing VLM-based VLA methods follow a common training paradigm: they first pretrain on large-scale robot datasets, and then fine-tune on new  settings for evaluation. Conducting large-scale co-training for every new small dataset and evaluation benchmark is extremely costly, which is why most works provide a high-performance base model that can be adapted with only a few task-specific demonstrations. In addtion, mixing them will lead to significantly inbalanced data ratios, necessitating careful tuning of sampling weights.
> >
> >
> > We understand the reviewer’s concern that the current test setting may not be fully straightforward. To further validate that HiMoE-VLA can effectively handle heterogeneity, we conducted an additional experiment: **from-scratch co-training on CALVIN ABC (EEF) and CALVIN D (Joint)**, compared with training only on CALVIN D (Joint). The results are summarized below:
> >
> > |              | CALVIN D(JOINT) | CALVIN ABC(EEF)+D(JOINT) |
> > | :----------: | :-------------: | :----------------------: |
> > |    $pi_0$    |      3.806      |    3.547(**0.259↓**)     |
> > | Ours w/o MoE |      3.819      |    3.777(**0.042↓**)     |
> > |     Full     |      3.826      |    4.012(**0.186↑**)     |
> >
> > [1] π0: A Vision-Language-Action Flow Model for General Robot Control
> >
> > [2] Fine-Tuning Vision-Language-Action Models: Optimizing Speed and Success
> >
> > [3] Unified Vision-Language-Action Model
> >
> > > **Q2: What about applying the proposed Hierarchical MoE to other domains? like images or languages. The proposed Hierarchical MoE looks general and can still make sense when action modality is not mentioned here.**
> >
> > Thank you for the question. Although our work focuses on robotic manipulation, the **core design principles of HiMoE are model-agnostic**, and in principle can be applied to other domains such as vision or language.
> >
> > Mixture-of-Experts architectures have shown strong potential in both language and vision. HiMoE could extend naturally to these domains as long as the model has access to a **global heterogeneity label**. For instance, in a multilingual language model, knowing the input language (e.g., English or Chinese) would allow an AS-MoE to specialize across languages, while an HB-MoE could further capture broader variations such as task type (translation, summarization, coding).
> >
> > Although we have not experimented with non-robotic domains in this paper, we believe the core idea—**decomposing heterogeneous factors across multiple levels**—is broadly applicable.
> >
> > ---
> >
> > Once again, thank you for questions and please let us know if you have any further questions.

---

> > > ### Author Response · Authors · 2025-11-27
> > > **A Polite Reminder**
> > >
> > > Dear Reviewer gPfg,
> > >
> > > I hope this message finds you well. As the discussion period is nearing its end, we wanted to ensure that we have addressed all your concerns satisfactorily.
> > >
> > > If there are any additional points or feedback you would like us to consider, please feel free to let us know. Your insights are invaluable to us, and we are eager to address any remaining issues to further improve our work.
> > >
> > > If you feel that our responses and the newly added results sufficiently address your concerns, we kindly ask you to consider updating your score accordingly.
> > >
> > > Thank you again for your time and effort in reviewing our paper.

---

### Official Review · Reviewer_8FHX · 2025-10-30

**Soundness:** 3
**Presentation:** 3
**Contribution:** 3
**Rating:** 6
**Confidence:** 4

**Summary:**

The paper presents a new VLA framework for handling heterogeneous robot data - including different action spaces as well as different embodiments and sensor configurations. The authors validate their method across a variety of simulated and real benchmarks and provide ablation studies to justify their design choices.

**Strengths:**

- The paper tackles an important problem of being able to learn robot policies from varies sources, including variations in embodiments and action spaces.
- The paper introduces two regularization components - one to handle variability in action spaces while the other handles knowledge sharing across embodiments.
- The authors conduct extensive experiments across both simulation and the real world to validate the usefulness of the proposed framework.
- The authors include detailed ablation studies to justify the design choices made in the proposed framework.

**Weaknesses:**

Including both weaknesses as well as questions tied to the weaknesses below.
- The authors mention being able to handle different sensor configurations (line 89). I am confused which of the experiments validate the variability in sensor configurations.
- It seems that the different action spaces are first projected into a unified vector representation, where each action space is consistently assigned to fixed positions within the vector. I am curious whether the network’s performance is sensitive to the dimensionality of each action space. In the current setup, the dimensions are relatively similar (7 for the single-arm setting and 14 for the bimanual case). However, if one were to train a single policy across embodiments with substantially higher-dimensional action spaces—such as 5-fingered hands or humanoids—would the same strategy still be effective? I would be interested to hear the authors’ perspective on this, acknowledging that it may be difficult to conduct such experiments within the rebuttal period.
- Missing citation in line 241.
- How many demonstrations are used for CALVIN and LIBERO? Also, does D->D on Calvin mean identical train and test settings? Further, in Tables 1 and 2, are all baselines finetuned on the same CALVIN and LIBERO datasets?
- The paper must discuss the limitations of the proposed method.

**Questions:**

It would be great if the authors could address questions in the weaknesses section.

---

> ### Author Response · Authors · 2025-11-23
> **Response to Reviewer 8FHX (1/2)**
>
> Thank you for the invaluable comments. Please find our responses below.
>
> > **W1: The authors mention being able to handle different sensor configurations (line 89). I am confused which of the experiments validate the variability in sensor configurations.**
>
> Thank you for the question. In our work, “different sensor configurations” refers to variations in camera setups across datasets, including differences in viewpoints (e.g., third-person vs. wrist-mounted cameras) and camera positions, which induce diverse observations of the scene. To validate that HiMoE effectively handles such heterogeneity, we analyzed expert activation patterns on CALVIN EEF, CALVIN Joint, and LIBERO EEF datasets. The results, shown in **Figures 4 and 5 in Appendix D** of the revised paper, indicate that CALVIN EEF and CALVIN Joint share similar expert activation patterns due to the same environment and camera settings, whereas LIBERO EEF exhibits a distinct distribution, reflecting its different camera configurations. These visualizations demonstrate that HiMoE can adapt its experts to variations in sensor setups, confirming the model’s ability to handle observation heterogeneity arising from different camera configurations.
>
> > **W2: It seems that the different action spaces are first projected into a unified vector representation, where each action space is consistently assigned to fixed positions within the vector. I am curious whether the network’s performance is sensitive to the dimensionality of each action space. In the current setup, the dimensions are relatively similar (7 for the single-arm setting and 14 for the bimanual case). However, if one were to train a single policy across embodiments with substantially higher-dimensional action spaces—such as 5-fingered hands or humanoids—would the same strategy still be effective? I would be interested to hear the authors’ perspective on this, acknowledging that it may be difficult to conduct such experiments within the rebuttal period.**
>
> This is a insightful question. In our current implementation, the model is not sensitive to the dimensionality of each action space due to our loss normalization strategy. In our method, we only supervise valid entries when computing the diffusion loss. Specifically, the  loss is computed **only on the valid action dimensions** for each data point, and the loss is normalized by the number of valid dimensions. Therefore, each data point contributes **equal weight**, regardless of whether its action space is 7-D, 14-D, or higher-dimensional.
>
> Because loss normalization is based on the valid dimensionality, the model does not inherently bias toward higher-dimensional action spaces. This strategy should remain effective when scaling to more complex embodiments (e.g., 5-finger dexterous hands or humanoids). Similar strategies have been successfully used in prior works [1] [2] including those on dexterous hand [3].
>
> [1] π0: A Vision-Language-Action Flow Model for General Robot Control
>
> [2] RDT-1B: a Diffusion Foundation Model for Bimanual Manipulation
>
> [3] Scalable Vision-Language-Action Model Pretraining for Robotic Manipulation with Real-Life Human Activity Videos
> > **W3: Missing citation in line 241.**
>
> Thank you for pointing this out. We have added the missing citation in the revised manuscript.

---

> > ### Author Response · Authors · 2025-11-23
> > **Response to Reviewer 8FHX (2/2)**
> >
> > > **W4: How many demonstrations are used for CALVIN and LIBERO? Also, does D->D on Calvin mean identical train and test settings? Further, in Tables 1 and 2, are all baselines finetuned on the same CALVIN and LIBERO datasets?**
> >
> > We used the **officially released demonstrations** for both CALVIN and LIBERO. Specifically, CALVIN D contains 4,764 episodes; LIBERO-10, LIBERO-Goal, LIBERO-Object, and LIBERO-Spatial include 379, 428, 454, and 432 episodes, respectively.
> >
> > In the **D→D** setting of CALVIN, the training and evaluation environments share the same overall layout. However, object locations are randomized during evaluation, and long-horizon execution makes each subtask state depend on the outcomes of previous subtasks. Therefore, D→D is used in our evaluation to test whether the model can adapt to new rollout conditions even when only limited data is available for that setting.
> >
> > For all baselines in Tables 1 and 2, the reported results are either taken directly from the original papers or obtained by finetuning the models on the same CALVIN and LIBERO datasets using the hyperparameter configurations recommended in their respective works. This ensures a fair and consistent comparison across methods.
> >
> > > **W5: The paper must discuss the limitations of the proposed method.**
> >
> > Thank you for the great suggestions. The following discussion of limitations have been added to Appendix E of the revised paper:
> >
> > Our current implementation feeds **all VLM layers’ features** into HiMoE and performs cross-attention using the full set of hidden-state key–value pairs. This creates two practical limitations. First, not all VLM layers necessarily contribute equally to downstream task execution, and treating all layers as equally important may introduce redundancy. Second, the hidden states include tokens from multiple viewpoints and text descriptions, yet the model currently injects them into HiMoE without distinguishing their relative importance. Adaptively filtering or weighing VLM features could further benefit the system, and exploring such mechanisms will be an important direction for our future work.
> >
> > Another limitation concerns data and model scale. Although our model performs well on the evaluated tasks, its overall scale is still modest relative to large vision–language foundation models. This limitation arises primarily from the limited availability of high-quality, diverse robotics datasets. The robotics datasets we use are much smaller and less diverse compared to the large-scale corpora used to train general VLMs. Increasing the model capacity, incorporating stronger pretrained VLMs, and training on more extensive and varied robotics datasets could further enhance generalization and robustness. Investigating how model scale interacts with embodiment-specific variability in learning will be an important topic for future research.
> >
> > ---
> >
> > Thank you once again for constructive feedback and please let us know if you have any further questions.

---

### Official Review · Reviewer_wPBu · 2025-10-31

**Soundness:** 3
**Presentation:** 3
**Contribution:** 2
**Rating:** 6
**Confidence:** 3

**Summary:**

The paper proposes a hierarchical mixture-of-experts (MoE) approach for a VLA framework aimed at robust transfer across diverse robot embodiments. To handle heterogeneous action spaces, two MoE modules—AC-MoE and HB-MoE—are introduced within a diffusion-based action module. For the vision–language component, a pretrained VLM is employed to extract intermediate key–value (KV) representations, which are fed into the action module. Experiments show that the proposed approach outperforms baselines on a range of manipulation tasks across four environments.

**Strengths:**

The work targets a generalist VLA policy applicable to multiple robot embodiments and environments, a direction with broad practical impact. The network design and the two specialized MoE modules are intuitive and well-motivated. Empirically, the method consistently outperforms strong baselines in two simulated and two real-world settings.

**Weaknesses:**

- The method appears sensitive to the choices of hyperparameters K and N (according to Table 6(c)).

- In the ablations, success rates are quite close. Considering the high variance over different runs, the incremental benefit of the proposed MoE modules seems to be marginal.

- Several details are unclear to me. Please refer to the questions.

**Questions:**

- HB-MoE appears very similar to DeepSeekMoE (Dai et al., 2024) in both architecture and training objective. What novel aspects distinguish HB-MoE from DeepSeekMoE?

- Is the specialization of MoE modules preferable to duplicating a single MoE module with the two regularization losses? Including this comparison in the ablation study would better support the hierarchical design choice.

- Some technical details are missing: How are heterogeneous actions encoded into a fixed-size vector? How are gating scores computed in the MoE modules (e.g., inputs, normalization, and temperature settings)?

---

> ### Author Response · Authors · 2025-11-23
> **Response to Reviewer wPBu (1/2)**
>
> Thank you for the valuable feedback, and we address the questions as follows.
>
> >  **W1: The method appears sensitive to the choices of hyperparameters K and N (according to Table 6(c)).**
>
> N and K are important hyperparameters in MoE models, as their combination directly affects the performance of MoE Model. Sensitivity to N and K is a common observation in previous MoE architectures (e.g., see Figure 3 of [1], Tables 2–3 of [2], Figures 8–9 of [3])
>
> In our method, the performance generally improves as N and K increase; see the results of N ranging from 2 to 32 and K from 2 to 4 in Table 9. For these configurations, all models achieve **higher performance than the “w/o MoE” baseline**, demonstrating the effectiveness of HiMoE. The performance drops when increasing K to 8. We hypothesize that this drop is due to **limited data**: as the model size grows while the dataset size remains fixed, the optimization process becomes more challenging, and the model is more prone to overfitting, leading to degraded performance.
>
> **References**
>
> [1] DeepSeekMoE: Towards Ultimate Expert Specialization in Mixture-of-Experts Language Models
>
> [2] LLaMA-MoE: Building Mixture-of-Experts from LLaMA with Continual Pre-training
>
> [3] From Sparse to Soft Mixtures of Experts
>
> > **W2: In the ablations, success rates are quite close. Considering the high variance over different runs, the incremental benefit of the proposed MoE modules seems to be marginal.**
>
> Thanks for your question. As shown in Table 9 of the revised manuscript, the default MoE configuration (**N = 32, K = 4**) achieves a success rate of **4.012**, which is notably higher than the **3.777** reported in Table 8 for the **"w/o MoE"** baseline. In Table 8, we used a smaller MoE configuration (**N = 8, K = 2**) to conduct ablation study for efficiency considerations, and this setting reaches **3.896** which, while shows an improvement over the baseline, is less striking.
>
> To address the reviewer’s concern, we re-ran the experiments in Table 8 using our **default (N = 32, K = 4)** setting. The updated results presented in the table below (and included in the revised paper) show larger success rate improvements in these ablations, furthering confirm the effectiveness of our MoE design.
>
> |   Setting   |     1     |     2     |     3     |     4     |     5     |   Sum.    |
> | :---------: | :-------: | :-------: | :-------: | :-------: | :-------: | :-------: |
> |   w/o MoE   |   0.918   |   0.837   |   0.744   |   0.681   |   0.597   |   3.777   |
> | Full-HB-MoE |   0.917   |   0.847   |   0.774   |   0.713   |   0.650   |   3.901   |
> | w/o AS-MoE  |   0.909   |   0.831   |   0.769   |   0.718   |   0.646   |   3.873   |
> | w/o HB-MoE  |   0.904   |   0.826   |   0.749   |   0.708   |   0.649   |   3.836   |
> |   w/o Reg   |   0.904   |   0.822   |   0.753   |   0.702   |   0.654   |   3.835   |
> |    Full     | **0.943** | **0.864** | **0.797** | **0.734** | **0.674** | **4.012** |

---

> ### Author Response · Authors · 2025-11-23
> **Response to Reviewer wPBu (2/2)**
>
> > **Q1: HB-MoE appears very similar to DeepSeekMoE (Dai et al., 2024) in both architecture and training objective. What novel aspects distinguish HB-MoE from DeepSeekMoE?**
>
> Thanks for the insightful comment. We indeed adopt the high-quality open-source DeepSeekMoE implementation, but HB-MoE differs from DeepSeekMoE in both the **design motivation** and **how MoE is integrated** into the model.
>
> **1. Different motivations and roles of MoE.**
>
> DeepSeekMoE replaces all Transformer FFNs with MoE blocks as universal processing units. In contrast, we apply HB-MoE only as a *heterogeneous feature extractor* and **only place MoE at the model boundaries** (i.e., near input and output ends). This design is specifically tailored for embodied data, where diverse action/state spaces require flexible specialization at the interface between heterogeneous data and the shared policy model.
>
> **2. Empirical evidence: full replacement degrades performance in embodied tasks.**
>
> We conducted experiments replacing *all* FFNs with DeepSeek-style MoE. After pretraining, we fine-tuned on CALVIN D (Joint). As shown below, fully replacing all FFNs significantly degrades performance, whereas our design delivers substantial improvements:
>
> |             |   1   |   2   |   3   |   4   |   5   | Sum.  |
> | :---------: | :---: | :---: | :---: | :---: | :---: | :---: |
> | Replace All | 0.926 | 0.820 | 0.716 | 0.648 | 0.563 | 3.673 |
> |    Ours     | 0.932 | 0.855 | 0.789 | 0.731 | 0.660 | 3.967 |
>
> This demonstrates that full MoE replacement, while effective for language models, is not suitable for embodied action learning.
>
> > **Q2: Is the specialization of MoE modules preferable to duplicating a single MoE module with the two regularization losses? Including this comparison in the ablation study would better support the hierarchical design choice.**
>
> Thank you for the valuable suggestion. We agree that duplicating a single MoE module with both regularization losses is a meaningful ablation study baseline, and we added this experiment. The results are shown below:
>
> |                              |   1   |   2   |   3   |   4   |   5   | Sum.  |
> | :--------------------------: | :---: | :---: | :---: | :---: | :---: | :---: |
> | single MoE with two reg loss | 0.914 | 0.839 | 0.757 | 0.688 | 0.615 | 3.813 |
> |           w/o Reg            | 0.904 | 0.822 | 0.753 | 0.702 | 0.654 | 3.835 |
>
> Interestingly, this variant performs even worse than the **"w/o Reg"** baseline. This indicates that forcing a **single** MoE module to learn **all heterogeneous factors simultaneously** is highly challenging, even with both regularization losses applied. Such entangled heterogeneity increases the difficulty of learning consistent routing patterns and influences the **convergence of the MoE**.
>
> In contrast, HiMoE’s hierarchical design—**first resolving action-space heterogeneity, then modeling other sources of variation**—provides clear specialization boundaries, enabling more effective expert learning on large-scale heterogeneous embodied data.
>
> > **Q3: Some technical details are missing: How are heterogeneous actions encoded into a fixed-size vector? How are gating scores computed in the MoE modules (e.g., inputs, normalization, and temperature settings)?**
>
> In our implementation, all heterogeneous actions/state are mapped into a **24-dimensional vector**, which is divided into three 8-dimensional segments:
>
> - **First 8 dims:** end-effector (EEF) actions
> - **Second 8 dims:** single-arm joint angles
> - **Third 8 dims:** joint angles of the second arm (used in bimanual settings)
>
> If a particular action type is not present in the dataset (e.g., no EEF actions), we zero-pad the corresponding segment. To distinguish between padded zeros and zero movement, we additionally concatenate a **24-dimensional boolean validity mask** that explicitly indicates which segments contain valid action information.
>
> For the MoE gating mechanism, each token’s **hidden state** is fed into a linear projection to compute the expert logits. These logits are then normalized using a **standard softmax** without temperature scaling, producing a normalized distribution over experts. The gate selects the **top-k** experts based on these softmax scores, and the selected probabilities are renormalized so that their weights sum to one. This design ensures stable routing and well-scaled mixture coefficients throughout training.
>
> These implementation details have been added in the **"Implementation Details"** subsection of the *Experiments* section in the revised paper.
>
> ---
>
> Once again, thank you for suggestions and please let us know if you have any further questions.

---

### Official Review · Reviewer_BLR6 · 2025-11-01

**Soundness:** 2
**Presentation:** 3
**Contribution:** 3
**Rating:** 4
**Confidence:** 5

**Summary:**

The paper targets a unified visuomotor-language policy that transfers across heterogeneous robot datasets with differing action spaces, embodiments, and sensors. It proposes HiMoE-VLA, a hierarchical MoE design with AS-MoE to handle discrepancies in action representations (e.g., joint vs. end-effector) and HB-MoE to balance broader cross-domain heterogeneity while retaining a shared transformer backbone; training combines flow-matching with specialization and balancing regularizers. Evaluations on CALVIN, LIBERO, and real-robot setups show consistent gains over prior VLAs.

**Strengths:**

- Tackles an important problem: transfer across diverse robot datasets and action spaces.
- Writing is generally clear; only minor phrasing/citation fixes needed.
- Strong results across sim and real benchmarks, outperforming existing VLAs.

**Weaknesses:**

- Core technical claims (e.g., non-transferability across action spaces) and architectural choice (MoE vs. simpler sharing/separate heads) aren’t fully justified (see questions).
- Limited insight into learned expert specialization/routing.
- Minor issues:
    - broken citation L249
    - CALVIN aggregation is sum and not avg as mentioned in the table headers.

**Questions:**

1. L60: Why are data from different action spaces “largely non-transferable”?
2. If the goal is to avoid cross-action sharing, why not use separate heads per action representation (or something like GR00T with an embedding indicating embodiment type as input)? If the goal is to share observation representations, why not fully share parameters (as in other VLAs)? Note this should be studied as an ablation of the method to avoid confounding factors from other sources like pre-training dataset, hyper-parameter tuning, etc.
3. What is the role of the shared expert in HB-MoE?
4. Can you provide expert routing analysis for AS-MoE and HB-MoE across datasets with different action spaces and observations?
5. What exactly is “MoE re-initialization during fine-tuning” (L240)?
6. In Table 6(b), are active parameters matched between “Full” and “w/o MoE,” or total parameters?
7. Will the codebase and dataset be open-sourced?
8. It is noteworthy that the ‘w/o MoE’ variant in Table 6(b) already outperforms all VLAs in Table 1. A breakdown of which elements of the training pipeline contributed most, perhaps in the appendix, would be valuable to the community.

---

> ### Author Response · Authors · 2025-11-23
> **Response to Reviewer BLR6 (1/4)**
>
> Thank you for the constructive feedback, and we address your questions as follows.
>
> > **W3: Minor issues**
>
> Thank you for pointing these out. We have corrected all minor issues in the revised version of the paper.
>
> > **Q1: L60: Why are data from different action spaces “largely non-transferable”?**
>
> Data from different action spaces are **largely non-transferable** because they represent different physical quantities and follow different kinematic structures:
>
> 1. **They describe different physical meanings.**
>
>    For example, **end-effector (EE) actions** describe the gripper’s Cartesian pose, while **joint-angle actions** describe individual joint rotations. They live in different coordinate systems and obey different constraints.
>
> 2. **Even within the same action type, robots may differ in DoFs and kinematic structures.**
>
>    A 6-DoF and a 7-DoF arm performing the **same** EE motion will produce **different** joint-angle trajectories because their kinematic structures differ. So, joint-space data are not directly aligned between embodiments.
>
> Due to the different meanings and embodiment-dependent kinematics, EE and joint-angle actions lie in **incompatible domains** and follow **different data distributions**. This mismatch limits how well knowledge learned from one action space can be transferred to another.
>
> To further verify this, we co-trained from scratch on CALVIN ABC (EE actions) and CALVIN D (joint-angle actions) using both $pi_0$ and our method (w/o MoE). We intentionally removed pretrained initialization to purely evaluate how well the model handles heterogeneous action spaces. Under these conditions, mixing EE and joint-angle data caused clear conflicts and hurt performance.
>
> |              | CALVIN D(JOINT) only | CALVIN ABC(EEF)+D(JOINT) |
> | :----------: | :------------------: | :----------------------: |
> |    $pi_0$    |        3.806         |    3.547(**0.259↓**)     |
> | Ours w/o MoE |        3.819         |    3.777(**0.042↓**)     |
>
> These results show that heterogeneous action spaces can negatively interfere with each other unless the model is specifically designed to handle them.

---

> > ### Author Response · Authors · 2025-11-23
> > **Response to Reviewer BLR6 (2/4)**
> >
> > > **Q2: If the goal is to avoid cross-action sharing, why not use separate heads per action representation (or something like GR00T with an embedding indicating embodiment type as input)? If the goal is to share observation representations, why not fully share parameters (as in other VLAs)? Note this should be studied as an ablation of the method to avoid confounding factors from other sources like pre-training dataset, hyper-parameter tuning, etc.**
> >
> > > **Q2.1: why not use separate heads per action representation (or something like GR00T with an embedding indicating embodiment type as input)?**
> >
> > First, we use an MoE architecture because approaches such as separate heads or embodiment-type embeddings require **manually specifying the number of action spaces or embodiments in advance**. This design becomes **inflexible** when incorporating new settings: every new action space or embodiment would **require adding a new input/output head or defining new embedding categories**, making the architecture difficult to scale. In addition, different action spaces may not be completely independent—there is still shared knowledge that can benefit learning. Separate heads, however, treat each action space as fully isolated and therefore cannot exploit this shared structure.
> >
> > In contrast, MoE provides **adaptive specialization** through its gating mechanism learned from data, allowing the model not only to select individual experts but also to form different **combinations of experts** based on the input distribution. This makes the architecture both **flexible and easily extensible**: when encountering a new action space or embodiment, the model can naturally adapt by activating **new combinations of existing experts**, without adding any new heads or modules. In addition, this mechanism allows the model to **share knowledge across action spaces when helpful, while keeping separation for space-specific features**, making MoE a much more efficient and scalable way to handle heterogeneous action data than relying on fixed, hand-designed mappings.
> >
> > For experimental validation, here we compare our method with two variants: 1: **Separate heads** for each action representation, and 2: **GR00T-style** model that takes an embodiment indicator as input. The results are summarized in the table below, and we have included this table in the revised version of the paper.
> >
> > |                |     1     |     2     |     3     |     4     |     5     |   Sum.    |
> > | :------------: | :-------: | :-------: | :-------: | :-------: | :-------: | :-------: |
> > | Separate heads |   0.914   |   0.833   |   0.753   |   0.696   |   0.631   |   3.827   |
> > |   GR00T-like   |   0.913   |   0.835   |   0.764   |   0.702   |   0.642   |   3.856   |
> > |  Our w/o MoE   |   0.918   |   0.837   |   0.744   |   0.681   |   0.597   |   3.777   |
> > |   Ours Full    | **0.943** | **0.864** | **0.797** | **0.734** | **0.674** | **4.012** |
> >
> > > **Q2.2: why not fully share parameters (as in other VLAs)?**
> >
> > Technically, our HiMoE-VLA **does** share all parameters: all experts are part of a single unified model, and the routing to experts is learned automatically under the supervision of our regularization loss. We don't manually assign any expert to any specific data source. We are not entirely sure what the reviewer meant by "**fully share parameters**". If this refers to a variant where all MoE modules are removed -- i.e., a fully dense model without expert routing -- the corresponding results can be found in Table 8 of the paper as "w/o MoE" (also included in the 3rd row of the table above). The performance drops significantly when our MoE design is removed.
> >
> > We are unsure if this satisfactorily addresses your question, particularly regarding the meaning of "fully share parameters". If not, we would greatly appreciate further clarification so that we can provide a more accurate response.

---

> ### Author Response · Authors · 2025-11-23
> **Response to Reviewer BLR6 (3/4)**
>
> > **Q3: What is the role of the shared expert in HB-MoE?**
>
> The shared expert captures **common knowledge that is agnostic to data heterogeneity**. This allows the other experts to focus solely on **input-specific variations** (e.g., sensor configurations) instead of repeatedly learning the same basic patterns. In this way, the shared expert enables **more efficient knowledge sharing** and stabilizes specialization among the remaining experts. And the shared expert is common in some former works, such as [1], [2].
>
> To further verify its effectiveness, we conducted an ablation study of **removing the shared expert**. The performance dropped moderaly, indicating that this component contributes to more stable specialization and overall stronger performance.
>
> |                   |   1   |   2   |   3   |   4   |   5   | Sum.  |
> | :---------------: | :---: | :---: | :---: | :---: | :---: | :---: |
> | w/o shared expert | 0.921 | 0.846 | 0.779 | 0.732 | 0.677 | 3.955 |
> |       Full        | 0.943 | 0.864 | 0.797 | 0.734 | 0.674 | 4.012 |
>
> [1] DeepSeekMoE: Towards Ultimate Expert Specialization in Mixture-of-Experts Language Models
>
> [2] LLaMA-MoE: Building Mixture-of-Experts from LLaMA with Continual Pre-training
>
> > **Q4: Can you provide expert routing analysis for AS-MoE and HB-MoE across datasets with different action spaces and observations?**
>
> Yes, and thank you for the suggestion. We evaluated the expert routing behavior on CALVIN Joint, CALVIN EEF, and LIBERO EEF datasets. The expert activation heatmaps for AS-MoE and HB-MoE are provided in Figure 4 and 5 in the Appendix D of revised version of paper.
>
> From Figure 4, we observe that CALVIN EEF and LIBERO EEF exhibit similar expert activation patterns, while CALVIN Joint shows a clearly different distribution, reflecting the differences in action space. From Figure 5, the expert activation patterns for CALVIN Joint and CALVIN EEF are similar, as these datasets share the same environment and observation settings except for the action space, whereas LIBERO EEF has a distinct activation pattern.
>
> These visualizations demonstrate that HiMoE effectively adapts its experts to handle heterogeneity across both action spaces and observations, confirming that the hierarchical MoE design enables specialized routing for heterogeneous data.
>
> > **Q5: What exactly is “MoE re-initialization during fine-tuning” (L240)?**
>
> "MoE re-initialization during fine-tuning" means that we **reset** MoE-related parameters with a Gaussian distribution at the beginning of fine-tuning. Our goal is for HiMoE to abstract feature into shared knowledge, but the **domain gap** between the pre-training data and the fine-tuninfg data is often quite large. In such cases, directly using the pretrained MoE as a feature extractor could be suboptimal.
>
> Please note that we only reinitialize the MoE parameters; all other transformer blocks within the action module, which were pretrained alongside and learned to cooperate with the MoE, are retained. This ensures that knowledge is effectively transferred from the HiMoE pretraining to the fine-tuning phase.
>
> In practice, we also found an alternative approach—**MoE-only warmup**, where we train only the MoE parameters for several steps before unfreezing the rest of the model. In this way, we can similarly mitigate the mismatch caused by large domain shifts. The results below are from fine-tuning on the CALVIN D joint data.
>
> |         |   1   |   2   |   3   |   4   |   5   | Sum.  |
> | :-----: | :---: | :---: | :---: | :---: | :---: | :---: |
> | re-init | 0.932 | 0.855 | 0.789 | 0.731 | 0.660 | 3.967 |
> | warmup  | 0.938 | 0.866 | 0.794 | 0.723 | 0.659 | 3.980 |

---

> > ### Author Response · Authors · 2025-11-23
> > **Response to Reviewer BLR6 (4/4)**
> >
> > > **Q6: In Table 6(b), are active parameters matched between “Full” and “w/o MoE”, or total parameters?**
> >
> > In the "Full" setting, we use an MoE with N = 32 total experts and a top-K routing of K = 4. This model has slighly more active parameters (4 more experts) than the "w/o MoE" baseline. We understand the reviewer’s concern that the observed performance gains might stem from the increased number of parameters rather than the proposed design itself.
> >
> > To address this, we conducted an additional experiment by increasing the parameter count of the "w/o MoE" model (achieved by enlarging the dimensions of token feature and FFN's intermediate size). We co-trained this model on CALVIN ABC (EEF actions) and CALVIN D (joint-angle actions) from scratch and the results are shown below:
> >
> > |           | Param(All \| Act.) |   1   |   2   |   3   |   4   |   5   | Sum.  |
> > | :-------: | :----------------: | :---: | :---: | :---: | :---: | :---: | :---: |
> > |  w/o MoE  |   3.238 \| 3.238   | 0.918 | 0.837 | 0.744 | 0.681 | 0.597 | 3.777 |
> > | add param |   4.101 \| 4.101   | 0.898 | 0.806 | 0.744 | 0.702 | 0.651 | 3.801 |
> > |   Full    |   4.068 \| 3.360   | 0.943 | 0.864 | 0.797 | 0.734 | 0.674 | 4.012 |
> >
> > The model with added parameters still largely underperforms the HiMoE model, despite having a **slightly larger total parameter count and significantly larger active parameter count during inference**. This demonstrates that the performance gains cannot be attributed solely to the number of parameters.
> >
> > Furthermore, the table below shows that the performance of "Full" and "w/o MoE" on CALVIN D(JOINT) is similar, whereas the performance diverges significantly when CALVIN ABC(EEF) is added. This further reinforces that HiMoE offers benefits beyond simply increasing the parameter count, particularly when training on heterogeneous action spaces.
> >
> >
> > |         | CALVIN D(JOINT) only | CALVIN ABC(EEF)+D(JOINT) |
> > | :-----: | :------------------: | :----------------------: |
> > | w/o MoE |        3.819         |    3.777(**0.042↓**)     |
> > |  Full   |        3.826         |    4.012(**0.186↑**)     |
> >
> > > **Q7: Will the codebase and dataset be open-sourced?**
> >
> > Yes. Beyond the core MoE implementation, our codebase also includes several additional components that are crucial for large-scale heterogeneous data training—for example, a multi-dataset training pipeline built on top of LeRobot that supports flexible weighting and scheduling across diverse datasets. We plan to release all relevant materials, including the full codebase, datasets, and model checkpoints. We hope these resources will benefit the community and further advance research on heterogeneous data learning.
> >
> > > **Q8: It is noteworthy that the ‘w/o MoE’ variant in Table 6(b) already outperforms all VLAs in Table 1. A breakdown of which elements of the training pipeline contributed most, perhaps in the appendix, would be valuable to the community.**
> >
> > First, the settings of **Table 1** and **Table 8** (corresponding to the previous Table 6(b) in the original submission) are fundamentally different and therefore not directly comparable.
> >
> > - **Table 1** reports results under *fine-tuning* on **CALVIN D (joint)**.
> > - **Table 8** reports *from-scratch co-training* on **CALVIN ABC (EEF)** + **CALVIN D (joint)**, which primarily evaluates the model’s capability to handle heterogeneous data.
> >
> > Because the goals and data differ, the performance numbers between the two tables should not be interpreted in relation to each other.
> >
> > Beyond the components discussed in the paper, our training pipeline includes two additional mechanisms that $pi_0$ does not support: **data mask** and **loss mask**.
> >
> > - **Data mask**: we concatenate the state vector with a boolean mask indicating which indices correspond to valid data versus padding.
> > - **Loss mask**: during training, instead of computing loss over the full action vector, we compute loss only over the valid dimensions of each action space.
> >
> > For the data mask and loss mask, we conduct ablation studies under the **w/o MoE** setting with from-scratch co-training on CALVIN ABC (EEF) + D (joint). The results are shown below:
> >
> > |                            |   1   |   2   |   3   |   4   |   5   | Sum.  |
> > | :------------------------: | :---: | :---: | :---: | :---: | :---: | :---: |
> > |        Base setting        | 0.918 | 0.837 | 0.744 | 0.681 | 0.597 | 3.777 |
> > |       w/o loss mask        | 0.896 | 0.800 | 0.726 | 0.667 | 0.607 | 3.696 |
> > | w/o datamask w/o loss mask | 0.886 | 0.797 | 0.714 | 0.649 | 0.580 | 3.626 |
> >
> >
> > These results provide a clearer breakdown of how each component of them contributes to the performance improvements observed in above table.
> >
> > ---
> >
> > Thank you once again and please let us know if you have any further questions.

---

> ### Author Response · Authors · 2025-11-27
> **A Polite Reminder**
>
> Dear Reviewer BLR6,
>
> I hope this message finds you well. As the discussion period is nearing its end, we wanted to ensure that we have addressed all your concerns satisfactorily.
>
> If there are any additional points or feedback you would like us to consider, please feel free to let us know. Your insights are invaluable to us, and we are eager to address any remaining issues to further improve our work.
>
> If you feel that our responses and the newly added results sufficiently address your concerns, we kindly ask you to consider updating your score accordingly.
>
> Thank you again for your time and effort in reviewing our paper.

---

### Author Response · Authors · 2025-11-23
**General Response**

We sincerely thank the Area Chair for their time and all the reviewers for their valuable and constructive feedback.

A revised version of the manuscript has been submitted to address the questions and suggestions raised. The key updates are summarized below:

**Experiments:**

- Added a more direct comparison evaluating the model’s ability to handle heterogeneous data (Table 7).
- Added comparisons with alternative approaches for handling heterogeneous data, including separate heads and GR00T-style embodiment embeddings (Table 6(b)).
- Expanded ablations using the default MoE configuration (N = 32, K = 4), including additional variants such as w/o HB-MoE and a single MoE with two regularization losses (Table 8).

**Expert Routing Visualizations:**

- Added AS-MoE and HB-MoE expert activation heatmaps across datasets with different action spaces and embodiments, illustrating how HiMoE adapts to heterogeneity (Appendix D).

**Clarifications:**

- Added implementation details explaining (i) how heterogeneous actions are encoded into fixed-size vectors. (ii) how gating scores are computed in MoE modules.

- Added a Limitations and Future Work section (Appendix E).

**Corrections & Presentation:**

- Fixed broken citation at L249 and corrected the CALVIN aggregation rule in the table headers.

- Revised Figure 1 to explicitly show the hierarchical flow—AS-MoE → HB-MoE → Transformer blocks → HB-MoE → AS-MoE—instead of a single opaque block, making the structure and expert interactions clearer in the overview.

Key revisions and newly added content are highlighted in blue in the manuscript. Sorry for the slow response due to the extensive experiments newly added. We sincerely appreciate the reviewers’ time and insightful comments, which helped us substantially improve the clarity and rigor of the paper. We are happy to provide any additional details if needed.

---

### Author Response · Authors · 2025-12-01
**Summary for Area Chairs**

Dear Area Chairs,

We sincerely thank you for stepping in during this unusual situation.

While we actively addressed all comments with extensive experiments and clarifications (including around **13** new experiments), it's very unfortunate that the abrupt suspension of the review platform prevented further dialogue from reviewers. We greatly appreciate your additional effort required to handle this situation and sincerely hope that you will evaluate the validity of our defense based on the scientific merits of our added evidence.

To address the reviewers' concerns, we provided clarifications and extensive supporting experiments. All revisions and new experimental results have been incorporated into the revised manuscript. The main responses are summarized below:

## **For Reviewer BLR6:**

For all technical weaknesses and questions raised by Reviewer BLR6, we have provided additional clarifications and new experimental results to support our responses. Given that the concerns raised are grounded in technical validations rather than subjective opinions, each point can be addressed through experiments, most of which are already outlined in the reviewer’s comments. Our added results consistently support the expected behavior, and we therefore believe that these clarifications and experimental results adequately address all of the reviewer’s concerns.

## **For Reviewer wPBu:**

Reviewer wPBu’s main concerns were:
(1) the method appears sensitive to the choices of hyperparameters K and N, (2) the incremental benefit of the proposed MoE modules seems marginal.

For concern (1): Our experimental results show a **general improvement** in performance as N and K increase, and the observed sensitivity is **expected**. Such behavior is **common** across MoE-based methods, including **DeepSeek-MoE** and **LLaMA-MoE**, making it a **reasonable characteristic** of this model family.

For concern (2): In response to the reviewer’s question about the effectiveness of our MoE modules, we provided **further clarification** and **updated** the ablation study using our default setting. The revised ablations **clearly demonstrate** the **effectiveness** of our MoE design.

The reviewer also suggested adding an additional ablation. We **appreciated** this suggestion, **incorporated** the requested ablation into our study, and provided an **analysis** of the results in the revised version.

## **For Reviewer 8FHX:**

Reviewer 8FHX provided **very positive remarks** on our work and noted that “it would be **great** if the authors could address questions in the weaknesses section.”

The reviewer’s primary concern was that no experiment explicitly validated the model’s ability to handle corresponding variability. To address this, we compared the MoE expert selections under different sensor configurations to **evaluate** the model’s robustness to such variability. The corresponding results have been added to **Appendix D.**

The reviewer also raised an **exploratory question** regarding the broader applicability of our method(W2). We responded by providing our **analysis and perspective** on why the method remains applicable and feasible under such extensions.

## **For Reviewer gPfg:**

First, we respectfully note that **several concerns** raised by Reviewer gPfg—such as W2 and Q2—do **not accurately reflect** the overall **quality** of our submission. Therefore, we believe that **a score of 4** based primarily on these points may not be fully **justified.**

The reviewer’s main concerns were:
(1) "the performance on the CALVIN benchmark was inferior to FLOWER", (2) "I cannot see any contributions in Figure 1."

For (1): We **respectfully disagree** that this should be considered a weakness of our method. We responded from **multiple perspectives**: our approach **outperforms** several strong and widely adopted **baselines**; the **release time** of FLOWER is very **close** to the submission of our work, meaning the two were developed independently and in parallel; and, most importantly, our method and FLOWER address **fundamentally different** problems.

For (2): We **clarified the intended purpose** of Figure 1 as a **high-level overview** rather than a **detailed presentation of contributions**. The contributions and architectural details of our HiMoE are clearly depicted in the main text and Figure 2. We also revised Figure 1 to include more details of the HiMoE action module in the updated manuscript.

The reviewer also asked why we did not co-train benchmarks together with large datasets before testing. We clarified why such an experimental setup is **not appropriate**, and **to directly address the underlying concern**, we conducted **additional experiments** that evaluating the model’s ability to handle heterogeneous data.

---

We hope that the extensive and constructive evidence we have provided will be given full consideration beyond the automatically frozen scores. Thank you again for your time and fair judgment.

---

### Meta-Review · Area_Chair_7mET · 2026-01-07

**Summary:**

The submission introduces HiMoE-VLA, a hierarchical MoE architecture for generalist vision-language-action policies, addressing heterogeneity in robot embodiments, action spaces, and sensors. Reviewers acknowledged the timeliness of the problem and strong empirical results on CALVIN, LIBERO, and real-robot transfers, outperforming prior VLA baselines. However, concerns included insufficient justification for MoE over simpler alternatives (e.g., separate heads, embodiment tokens), limited insight into expert specialization/routing, hyperparameter sensitivity, and marginal contributions.

The authors submitted a major revision with extensive new experiments, ablations, visualizations, and clarifications, but a platform accident prevented reviewers from fully engaging and reassessing the new evidences. To the AC, the need for such substantial post-submission revisions itself indicates the original work was not yet ready for publication. On the other hand, while the AC sides with the authors that the lack of direct empirical comparison to concurrent work (FLOWER) is not necessarily fair, the rebuttal reveals earlier work which the authors are aware of (e.g. Unified Vision-Language-Action Model, from June 2025) was not mentioned or compared, suggesting room for more comprehensive comparisons.

Overall, the reviewers showed moderate enthusiasm without strong championship, which is unfortunately required for acceptance when calibrating papers this year. The authors are encouraged to improve and resubmit this work to future venues.

**Reviewer Concerns:**

Mitigated or partially addressed by rebuttal:

- Justification for MoE vs. alternatives, role of shared expert, routing analysis, parameter matching: New ablations (Tables 6b, 7, 8), visualizations (Appendix D), and clarifications provided.
- Hyperparameter sensitivity: Explained as typical MoE behavior; additional ablations at default settings added.
- Sensor variability handling: New routing analysis across configurations (Appendix D).
- Empirical comparison to FLOWER and assessment of contributions: Authors argued different scope and parallel development; no direct comparison added.

**Reviewer Scores:**

- BLR6: Original 4 → likely increasing.
- wPBu: Original 6 → likely remain.
- 8FHX: Original 6 → likely remain.
- gPfg: Original 4 → likely unchanged (authors dispute validity rather than fully rebut on merits).

---

### Decision · Program_Chairs · 2026-01-26

Reject